# Human NMD ensues independently of stable ribosome stalling

Evangelos D. Karousis [1], Lukas-Adrian Gurzeler [1,2], Giuditta Annibaldis[1], René Dreos[3] & Oliver Mühlemann [1✉]

Nonsense-mediated mRNA decay (NMD) is a translation-dependent RNA degradation pathway that is important for the elimination of faulty, and the regulation of normal, mRNAs. The molecular details of the early steps in NMD are not fully understood but previous work suggests that NMD activation occurs as a consequence of ribosome stalling at the termination codon (TC). To test this hypothesis, we established an in vitro translation-coupled toeprinting assay based on lysates from human cells that allows monitoring of ribosome occupancy at the TC of reporter mRNAs. In contrast to the prevailing NMD model, our in vitro system reveals similar ribosomal occupancy at the stop codons of NMD-sensitive and NMD-insensitive reporter mRNAs. Moreover, ribosome profiling reveals a similar density of ribosomes at the TC of endogenous NMD-sensitive and NMD-insensitive mRNAs in vivo. Together, these data show that NMD activation is not accompanied by stable stalling of ribosomes at TCs.

[1] Department of Chemistry and Biochemistry, University of Bern, CH-3012 Bern, Switzerland. [2] Graduate School for Cellular and Biomedical Sciences, University of Bern, CH-3012 Bern, Switzerland. [3] Center for Integrative Genomics, Université de Lausanne, CH-1015 Lausanne, Switzerland. ✉email: oliver.muehlemann@dcb.unibe.ch

Nonsense-mediated mRNA decay (NMD) is a translation-dependent mRNA surveillance pathway that targets physiological as well as aberrant mRNAs for degradation. Initially NMD was considered to be a quality control pathway that degrades mRNAs harboring a premature termination codon (PTC) within their open reading frame (ORF). These PTCs could originate from mutations as well as transcriptional or splicing errors[1]. The fact that NMD targets endogenous mRNAs that encode functional proteins revealed an important role of NMD as a post-transcriptional gene expression regulation mechanism, affecting a series of biological functions ranging from tissue differentiation to protection of host cells from RNA viruses (reviewed in refs. [2–6]).

Since NMD is crucial for a wide range of biological functions, an accurate and highly specific recognition mechanism of mRNAs that need to be engaged in the pathway is essential. Many studies have addressed the early events of NMD activation and overall it seems that the NMD machinery can "sense" many different cues from each mRNP. The position of splicing sites relative to the termination codon (TC) through the corresponding exon junction complexes (EJCs), the length of the 3′UTR, the presence of specific protein factors and the dynamics of translation termination are features that have been directly linked to the sensitivity of mRNAs to NMD. This surveillance mechanism is orchestrated through the omnipresent RNA helicase UPF1, a universal NMD factor that is crucial for NMD activation through phosphorylation (reviewed in refs. [7,8]).

An important feature of NMD is that it is dependent on translation, either on the first or a later round[9–12]. Evidence supporting the formation of complexes containing NMD and translation termination factors suggested a functional connection between the two processes[13–17]. In view of this data, a currently prevailing working model for NMD activation suggests that NMD ensues when translation termination is aberrant, either because it requires additional factors for ribosome release or because it is not fast enough. Based on evidence from S.cerevisiae extracts and rabbit reticulocyte lysate (RRL), it was proposed that ribosomes stall at NMD-triggering TCs, suggesting a kinetically slower translation termination that may be attributed to the absence of the poly(A)-binding protein (PABPC1 in mammals, Pab1 in yeast) in the vicinity of the TC[18,19]. PABPC1 can antagonize NMD when tethered on NMD-sensitive mRNA reporters and in vitro, PABPC1 competes with UPF1 for binding eRF3[20,21]. These data entertain the hypothesis that the difference between productive and aberrant, NMD-eliciting translation termination might rely on whether PABPC1 or UPF1 interacts with eRF3 at the terminating ribosome, thereby either promoting or inhibiting translation termination[22]. It should be noted, however, that PABP has been shown to stimulate translation termination only under non-physiological, limiting concentrations of release factors using a eukaryotic reconstituted system[23]. Using a similar approach, it was suggested that under decreased concentrations of release factors, the conserved NMD factor UPF3B can delay translation termination and dissociate post-termination ribosomal complexes that are devoid of a nascent peptide[16]. This finding and additional evidence documenting that the eRF3-interacting C-terminal domain of PABPC1 is not required for its NMD antagonizing capacity[4,24,25] suggest that the "UPF1 versus PABPC1 competition model" is oversimplified.

A better understanding of the dynamics of translation termination in the context of NMD-sensitive mRNAs is important to comprehend the activation of the NMD pathway. Previous approaches in measuring ribosomal density at the TC of reconstituted mammalian in vitro translation systems have offered important insights into the roles of crucial translation termination factors[26,27]. However, the fact that they are developed based on short open reading frames and that the reconstituted systems lack essential NMD factors limits their application in understanding translation termination in the context of NMD.

Here we present the development of an in vitro biochemical approach that allows the detection and comparison of ribosomal density at the TC using human cell lysates and in vitro transcribed mRNAs. We show that in this system, the occupancy of ribosomes at the TC is similar on both NMD-sensitive and insensitive reporter mRNAs, as well as in the presence or absence of a poly(A) tail. We complement these results by comparing the ribosomal density at the TC of mRNAs in vivo by ribosome profiling, which also revealed a similar occupancy of ribosomes at TCs, independently of their sensitivity to NMD.

## Results

**Assessment of ribosomal density at the TC of in vitro translated mRNAs in human cell lysates.** Previous work has suggested that prolonged ribosomal occupancy at the TC is a characteristic feature and maybe even the trigger for an mRNA to be subjected to the NMD pathway[18,19]. In order to assess ribosomal density at the TC, we developed an in vitro assay that allowed us to reproducibly identify terminating ribosomes on in vitro transcribed reporter mRNAs, based on previous protocols for the production of translation-competent cell lysates[28]. We opted for an approach that would allow all stages of translation of different mRNAs to occur using HeLa cell lysates followed by toeprinting assays. To this end, Micrococcal nuclease (MNase)-treated HeLa cell extracts were incubated with in vitro transcribed and capped mRNA reporters that harbor a humanized Renilla luciferase (Rluc) ORF followed by a 200 nucleotide long 3′UTR and an 80 nucleotides long poly(A) tail (reporter 200+ pA) (Fig. 1a). After optimization of different parameters such as mRNA concentration, incubation time, and titration of magnesium concentration to ensure efficient translation activity (Supplementary Fig. 1), we utilized our in vitro translation-competent lysates to assess ribosome occupancy at the TC of reporter mRNAs by toeprinting assay. After a 50-min incubation at 33 °C, robust Rluc activity was measured, documenting the translation competence of the lysate (Fig. 1b). No luminescence, and hence translation activity, was detected in the presence of the translation inhibitor puromycin. The in vitro translation reaction was followed by a primer extension reaction (toeprint assay) using a 5′ $^{32}$P-labeled oligonucleotide that binds the reporter mRNA 66 nucleotides downstream of the Rluc TC, which was optimized to yield signals originating from Rluc TC region (Fig. 1a, red arrow). After purification, the cDNA molecules were analysed by denaturing polyacrylamide gel electrophoresis and autoradiography. A puromycin-treated control reaction was used to distinguish between translation-dependent toeprints and translation-independent break-offs of the reverse transcriptase. To observe reproducible translation-dependent toeprints originating from ribosomes at TCs, we titrated a broad range of biochemical parameters. Among these, the appropriate $Mg^{2+}$ concentration turned out to be crucial to stabilize ribosomes on mRNA after translation, in agreement with previous observations[29]. The optimization of the $Mg^{2+}$ concentration improved the sensitivity of the translation-dependent toeprints and enabled the use of our protocol with lysates produced from smaller scale cell cultures ($<4 \times 10^7$ cells), which are more challenging to generate due to their small volume (Supplementary Fig. 1). This technical improvement allowed us, for example, to produce lysates from cells with an siRNA-mediated knockdown. As shown in Fig. 1c, a translation-dependent band appeared 18 nucleotides downstream of the first nucleotide of the TC in transcript 200+ pA. To distinguish whether translation-dependent bands correspond to

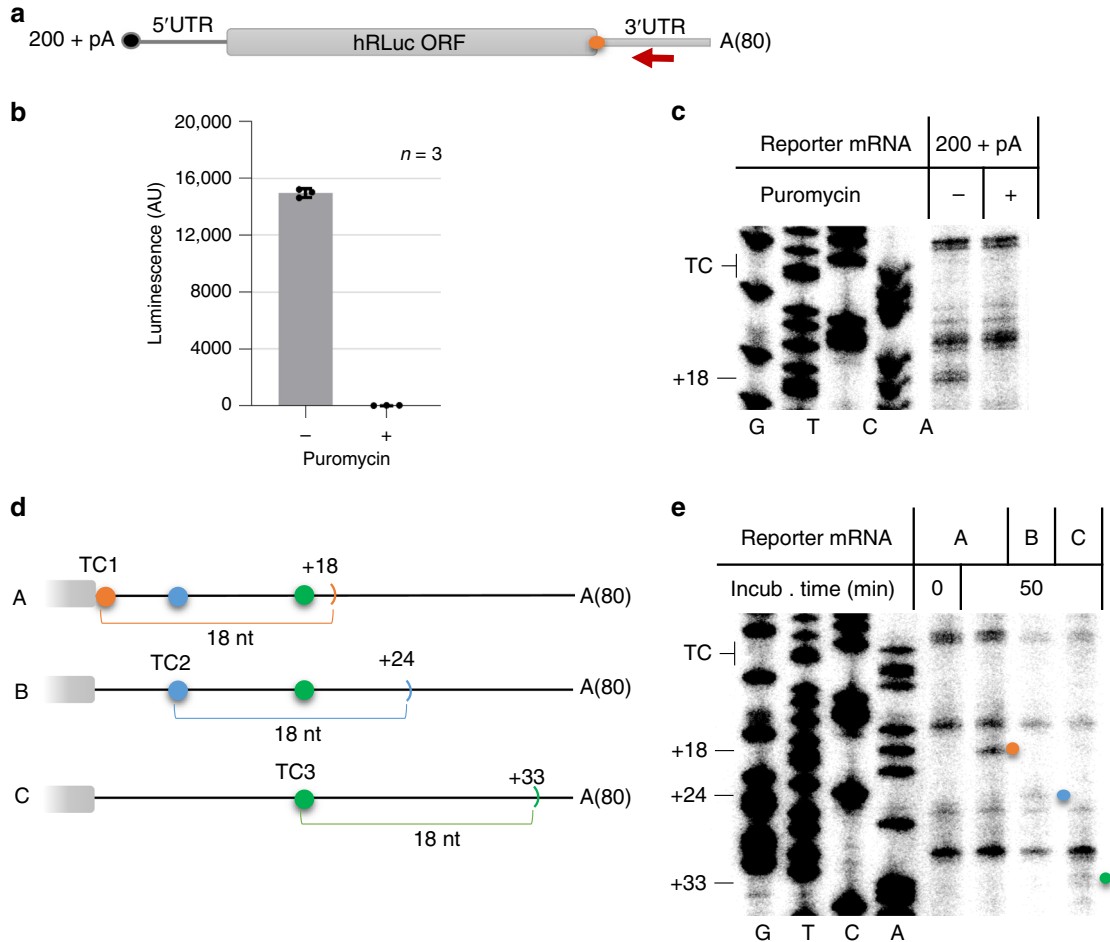

**Fig. 1 Toeprint analysis allows the detection of ribosomes at the termination codon of in vitro translated reporter mRNAs. a** Schematic representation of the in vitro synthesized capped (black dot) and polyadenylated reporter mRNA coding for humanized *Renilla* Luciferase (hRluc). The red arrow denotes the radiolabeled toeprint primer binding 66 nts downstream of the TC (orange dot). The 200 nt-long 3′UTR is followed by an 80 nts long template-encoded poly(A) tail depicted as A(80). **b** Rluc activity measurements of in vitro translation reactions. Luminescence is depicted as arbitrary units (AU), mean values ± standard deviations of 3 technical replicates are shown, values of the individual measurements are indicated by dots. **c** Toeprint analysis with the 200 + pA reporter transcript shown in **a**. Translation was performed in HeLa lysates for 50 min in the presence or absence of puromycin. To locate the positions of the toeprints, a Sanger sequencing reaction was run in parallel (G, T, C, A) using the same primer. The positions of termination codon (TC) and the toeprint band 18 nts downstream of the first nucleotide of the TC (+18), which originates from ribosomes located at the TC, are indicated. **d** Schematic representation of in vitro synthesized Rluc reporter transcripts A (=200 + pA), B and C, which differ with regards to the position of the termination codon (TC1, TC2, and TC3, respectively). The expected position of toeprints originating from ribosomes at the corresponding TC relative to the original position of TC1 is shown (+18, +24, and +33, respectively). **e** Toeprint analysis with the reporter transcripts shown in **d**. The translation-dependent bands corresponding to toeprints from ribosomes at the respective TCs are marked with dots following color code of **d**. In vitro translation and toeprint analysis were performed as in **c**. Source data are provided as a Source Data File.

ribosomes preventing the reverse transcriptase from passing or whether they originate from mRNA molecules cleaved during translation, we phenol-extracted the mRNA molecules prior to the primer extension step. Phenol extraction or heating to 95 °C of the reactions after translation leads to the disappearance of translation-dependent bands that originate from ribosomes, whereas bands originating from cleaved mRNAs will persist. While such cleaved mRNA fragments were detected in a series of toeprint experiments performed in rabbit reticulocyte lysates (Supplementary Fig. 2), the +18 translation-specific band disappeared under denaturing conditions in HeLa lysates, excluding the possibility that this band represents a co-translational RNA cleavage event (Supplementary Fig. 2). In order to further verify that the translation-dependent +18 bands indeed derived from ribosomes residing at the TC, we designed reporter transcripts in which we moved the position of the TC. Moving the TC 6 or 15 nucleotides downstream of the original position resulted in a

corresponding shift of the translation-dependent bands, confirming that our assay reliably detects toeprints originating from ribosomes at stop codons (Fig. 1E).

**Detection of stable ribosome stalling using toeprint assays.** To further assess whether our toeprint assay is suitable to monitor changes of ribosomal occupancy at the TC, we introduced the regulatory peptide of the human cytomegalovirus (hCMV) gp48 upstream open reading frame 2 (uORF2)[30] into the 200 + pA reporter construct. Translation of the hCMV uORF2 peptide inhibits termination at the TC, because the presence of a Pro-tRNA at the P site and a TC at the A site cause ribosome stalling[31]. We in vitro transcribed two versions of the modified 200 + pA reporter construct, one appending the hCMV stalling peptide sequence to the Rluc ORF and the other harboring a mutation that leads to the recruitment of an Ala-tRNA (GCG codon) instead of a Pro-tRNA (CCU codon) at the P-site of the

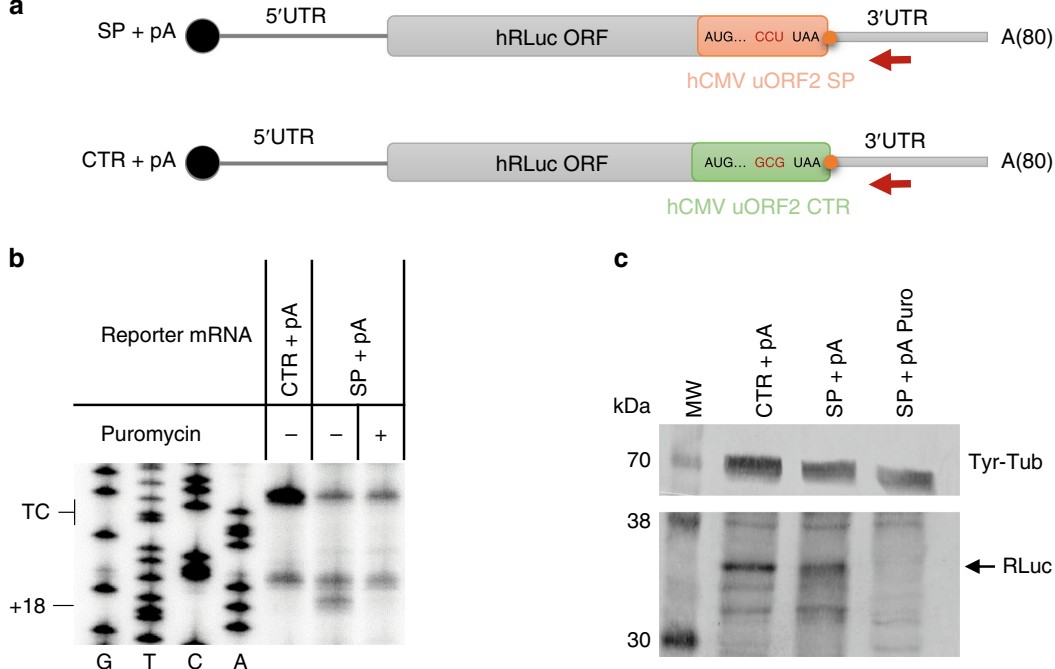

**Fig. 2 Stable ribosome stalling caused by the addition of the hCMV uORF2 stalling peptide downstream of the Rluc ORF. a** Schematic representation of modified RLuc reporter mRNA constructs containing the hCMV uORF2 stalling peptide (SP + pA) or a non-stalling control sequence (CTR + pA). The red arrow denotes the radiolabeled toeprint primer and the orange dot the termination codon. The 200-nt long 3′UTR is followed by an 80 nt-long poly(A) tail depicted as A(80). **b** Toeprint analysis of ribosome occupancy at the TC of the hCMV uORF2 was performed after in vitro translation in HeLa cell lysates for 50 min at 33 °C of equimolar amounts of the two reporter mRNAs described in **a**. Sanger sequencing reactions were run in parallel (G, T, C, A) to locate the positions of the toeprints. The position of the TC and the toeprint band corresponding to the ribosomes at the TC (+18) are indicated. **c** Representative western blot analysis of an aliquot of the translation reactions analyzed in **b**. Equal amounts of the translation reactions were separated on a 12% SDS-PAGE, transferred onto a nitrocellulose membrane and probed for tyrosine-tubulin (Tyr-Tub) and Rluc protein. The band corresponding to full-length Rluc protein is denoted by an arrow. The experiment was repeated three times. Source data are provided as a Source Data File.

terminating ribosome (Fig. 2a). The second reporter transcript served as a control, as it does not lead to ribosome stalling. The two reporter mRNAs were translated in HeLa cell lysates and were subjected to toeprint analysis as described before. While no (or only an extremely faint) toeprint at the +18 position could be detected with the non-stalling control transcript (CTR + pA), the transcript with the stalling peptide (SP + pA) showed a robust translation-dependent +18 band, demonstrating that our assay is suitable to measure differences in ribosomal occupancy at TCs (Fig. 2b). Since the C-terminal extension of the Rluc ORF by the hCMV ORF2 sequence inactivated the luciferase enzyme, translation of the reporter transcripts had to be verified by western blot for this experiment (Fig. 2c). Overall, the above results verified that our system can reproducibly detect terminating ribosomes as well as differences in ribosomal density at the termination codon.

**Depletion of the recycling factor ABCE1 leads to increased ribosome occupancy at the TC.** Next, we wanted to assess whether our toeprint assay is also able to identify changes of ribosome density at the TC under aberrant translation termination conditions. A previous study showed that reduction of the eukaryotic recycling factor ABCE1 can lead to prolonged ribosome occupancy at the TC[27]. In order to address whether we can recapitulate this event in our assay, we prepared lysates from siRNA-mediated control and ABCE1 knockdown cells (ABCE1 KD) and compared ribosomal density by toeprint assays. The efficacy of the ABCE1 knockdown was documented by probing equal amounts of lysates by western blotting (Fig. 3a). The higher intensity of the +18 band in the ABCE1-depleted lysate indicates an increased ribosome occupancy at the TC in the absence of the

recycling factor (Fig. 3b). This confirms that our in vitro system has the capacity to monitor differences in ribosome occupancy at the TC after depleting translation termination-related factors.

**NMD sensitivity of reporters with different 3′UTR lengths.** To directly test the prevailing NMD model by assessing whether NMD-sensitive mRNAs induce ribosome stalling at the TC, we designed a series of reporter mRNAs that differ in the length of the 3′UTR and portray different sensitivities to NMD. The reporters were designed in a way that allows the comparison of ribosome occupancy at the TC in toeprint assays using the same radiolabeled primer. All reporters share an identical 5′UTR, a humanized Rluc ORF and an identical 200 nucleotide long 3′UTR (construct 200 + pA). In addition to this sequence, a second construct has the 3′UTR extended to 1400 nucleotides (1400 + pA; Fig. 4a). The two constructs were transiently transfected into HeLa cells that additionally express a TCR-β NMD reporter and to test if they were targeted by NMD, we knocked down the key NMD factor UPF1. The efficacy of the UPF1 knockdown was monitored by western blotting (Fig. 4b) and the steady-state mRNA levels of the Rluc reporters 200 + pA and 1400 + pA were assessed by RT-qPCR (Fig. 4c). While the relative abundance of the 200 + pA transcript remained almost unchanged upon UPF1 depletion, the level of the 1400 + pA increased 4-fold, indicating that the transcript with the long 3′UTR (1400 + pA) was a target for NMD, whereas the one with the short 3′UTR (200 + pA) was not. This observation is in agreement with previous reports showing that long 3′UTRs can render mRNAs sensitive to NMD[32–35]. The mRNA levels of the TCR-β NMD reporter and the endogenous NMD-sensitive Retinitis Pigmentosa 9

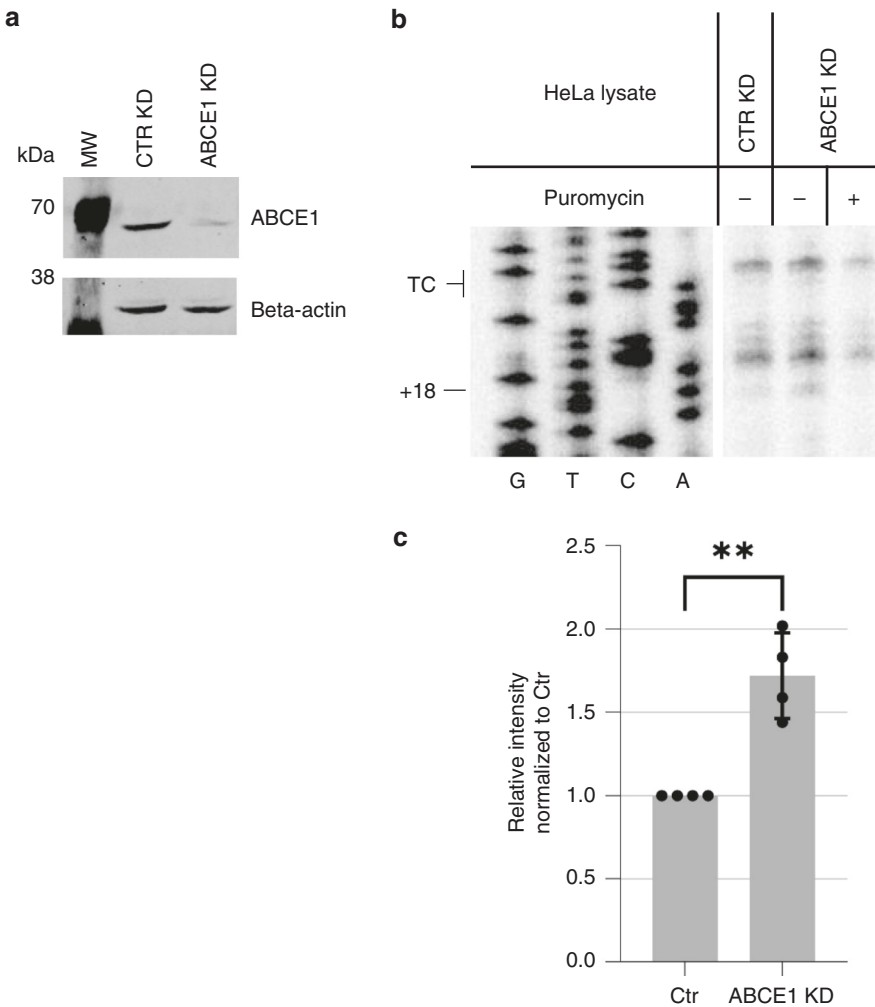

**Fig. 3 Increased ribosome occupancy at the TC of in vitro translated mRNAs under decreased concentrations of the recycling factor ABCE1. a** Western blot analysis of ABCE1 depletion in HeLa lysates. Cell lysates equivalent to $4 \times 10^5$ cells treated with a CTR or ABCE1 siRNAs were loaded on a 10% SDS-PAGE, transferred onto a nitrocellulose membrane and probed for ABCE1 and beta-actin. **b** Toeprint analysis of ribosome occupancy at the TC under ABCE1 depletion was performed after in vitro translation of the 200 + pA construct for 50 min at 33 °C in lysates from cells treated with CTR or ABCE1 siRNAs. Sanger sequencing reactions were run in parallel (G, T, C, A) to locate the positions of the toeprints. The position of the TC and the toeprint band corresponding to the ribosomes at the TC (+18) are indicated. **c** Relative quantification of the translation-dependent +18 band under ABCE1 conditions of 4 independent toeprinting experiments, normalized to the translation-independent mRNA signals and to the values of the Ctr KD conditions. Mean values ± standard deviation are shown and the values of the individual experiments are depicted by dots (Unpaired, two-sided statistical *t*-test: Ctr/ABCE1 *p* value: 0,0014). Source data are provided as a Source Data File.

Pseudogene transcript (RP9P)[35] confirmed that NMD activity was reduced in all UPF1 KD samples (Supplementary Fig. 3).

**Similar ribosome density at NMD-sensitive and insensitive TC.** It has been proposed that a long physical distance between the TC and the poly(A) binding protein (PABP) may hinder translation termination by causing prolonged ribosomal occupancy at the TC[18,19]. Previous reports have also suggested that PABP can facilitate translation termination under decreased release factors concentrations in vitro[23] and we therefore wanted to assess in our toeprint assay whether ribosome occupancy at the TC is affected by the presence or absence of a poly(A) tail, or by its physical distance from the TC. For this reason, the toeprint assay was performed using both NMD-sensitive (with the 1400 nucleotides long 3′UTR) and insensitive reporter mRNAs (with 200 nucleotides long 3′UTR) in two variants, either containing an 80 nucleotides long poly(A) tail or lacking it (Fig. 5a). To this end, equimolar amounts of the in vitro transcribed, capped and

purified reporter mRNAs were used for translation in HeLa lysates as described above, followed by primer extension reactions. Again, puromycin-treated samples were run alongside to unambiguously identify the translation-dependent toeprints at position +18 relative to the TC (Fig. 5b, lanes 2, 4, 6, and 8). The intensity of the +18 band was similar for the 200 + pA and the 1400 + pA transcripts, indicating that ribosome occupancy at the TC was not affected by the 3′UTR length (Compare lanes 1 and 5). Furthermore, we could also not detect a significant difference in the intensity of the +18 band depending on whether or not the transcripts contained a poly(A) tail, suggesting that, at least in this in vitro system, the presence of a poly(A) tail in the vicinity of the TC has no termination-promoting effect (Compare lanes 1 with 3 and 5 with 7). Altogether, this result is inconsistent with the prevailing NMD model proposing prolonged stalling at NMD-eliciting TCs. However, it is also possible that our in vitro toeprinting assay may simply not fully recapitulate the situation in vivo.

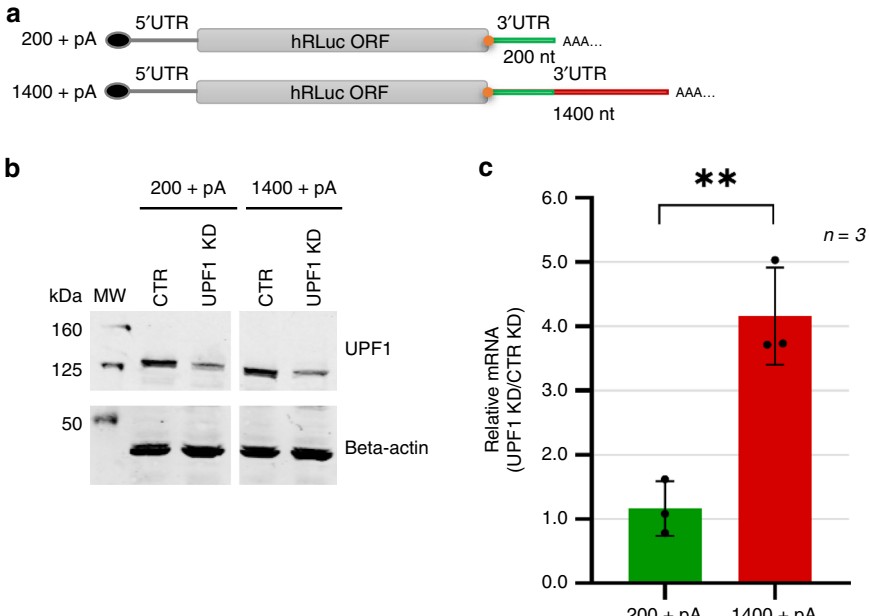

**Fig. 4 Extension of the 3′UTR renders the Rluc reporter mRNA NMD-sensitive in vivo. a** Schematic representation of Rluc reporter constructs with either a 200 nts long (200 + pA) or a 1400 nt-long 3′UTR (1400 + pA), modified for expression in human cells. The first 200 nts of the 3′UTR (green line) are identical in both constructs, the additional 1200 nts (red line) of the 1400 + pA 3′UTR correspond to a head-to-tail duplicated sequence originating from the ampicillin resistance gene. The constructs were cloned into pcDNA3.1(−) expression plasmid, where their expression is controlled by a CMV promoter and a bovine growth hormone polyadenylation signal. **b** Western blot analysis to monitor UPF1 knockdown efficacy in HeLa cells transiently expressing either 200 + pA or 1400 + pA Rluc reporter mRNA. Cell lysates equivalent to $2 \times 10^5$ cells were loaded on a 10% SDS-PAGE, transferred onto a nitrocellulose membrane and probed for UPF1 and beta-actin. **c** Relative abundance of 200 + pA and 1400 + pA mRNAs in cells depleted for UPF1 (UPF1 KD) normalized to cells with a control knockdown (CTR KD) were measured by RT-qPCR. Mean values ± standard deviations of 3 biological replicates are shown, values of the individual experiments are indicated by dots. (Unpaired, two-sided statistical *t*-test: 1400 + pA vs 200 + pA *p* value: 0.0039). Source data are provided as a Source Data File.

To test in vivo whether ribosomes show an increased occupancy at TCs of NMD-triggering mRNAs, we analysed ribosome profiling data to assess ribosome occupancy at the TC between NMD-sensitive and NMD-insensitive transcripts originating from previous work in the lab that was also performed in HeLa cells[36]. The ribosome profiling protocol was modified to assess specifically ribosome occupancy at the TC of endogenous mRNAs by omitting cycloheximide and applying instead snap-freezing of the cell lysates in liquid nitrogen[37]. Metagene analysis of ribosome-associated footprints from three independent experiments showed that under these conditions, ribosomes tend to reside longer on average at the TC than at a codon within the ORF (Fig. 5d, top). A list of 678 NMD-sensitive transcripts was compiled by including the most abundant isoform for each of previously identified NMD-sensitive genes that is stabilized under inhibited NMD (UPF1 KD) conditions[35]. When all 60'000 transcripts were ordered according to their ribosome occupancy at the TC (Fig. 5d, heatmap) and then labeled if they belong to the set endogenous NMD-sensitive transcripts (NMD targets), it became apparent that NMD-sensitive transcripts were not enriched among the transcripts with high ribosome occupancy at the termination codon. To normalize for the overall translation efficiency of a given mRNA, we determined the relative ribosome occupancy at TCs compared to CDS for each transcript by defining the ratio of the total counts of ribosome-derived reads aligning to the TC relative to the average of ribosome reads mapping at the CDS. In order to assess whether ribosomes reside longer at the TCs of NMD-sensitive mRNAs, we compared this relative stop codon ribosome occupancy between NMD-sensitive and insensitive mRNAs (Fig. 5e) and found no statistically significant difference between NMD-sensitive and insensitive

mRNAs. This result corroborates our in vitro data and altogether challenges the view that ribosome stalling at the TC is a hallmark of NMD-sensitive mRNAs.

## Discussion

We report here the development of an in vitro assay to examine the ribosome occupancy at the TC of NMD-sensitive and NMD-insensitive mRNAs using translation-competent lysates from human cells. In contrast to previous evidence originating from in vitro studies performed with yeast extracts and rabbit reticulocyte lysate[18,19], we found a similar ribosome occupancy at the TC of reporter mRNAs, independently of whether they contained a long 3′UTR that renders them sensitive to NMD in vivo. In addition, omitting a poly(A) tail from our reporter constructs did not affect the ribosome occupancy at the TC. These results from our in vitro toeprinting system are corroborated by our ribosome profiling experiments, in which normalized ribosome occupancy at TCs of mRNAs did not correlate with whether an mRNA was a target for the NMD pathway or not. Thus, our in vitro and in vivo data are in disagreement with a suggested working model for NMD, which posits that inefficient or aberrant translation termination that can be detected as ribosome stalling at the TC is the signal for NMD to ensue[18,38].

In vitro translation systems have significantly contributed to assess a broad range of translation-related mechanisms[39–41]. However, the study of translation termination in vitro using mammalian systems has remained technically challenging and has only recently gained broader attention. The cryo-electron microscopic analysis of ribosomal complexes isolated from in vitro translation reactions in rabbit reticulocyte lysate yielded structural information about distinct steps of translation[42], and

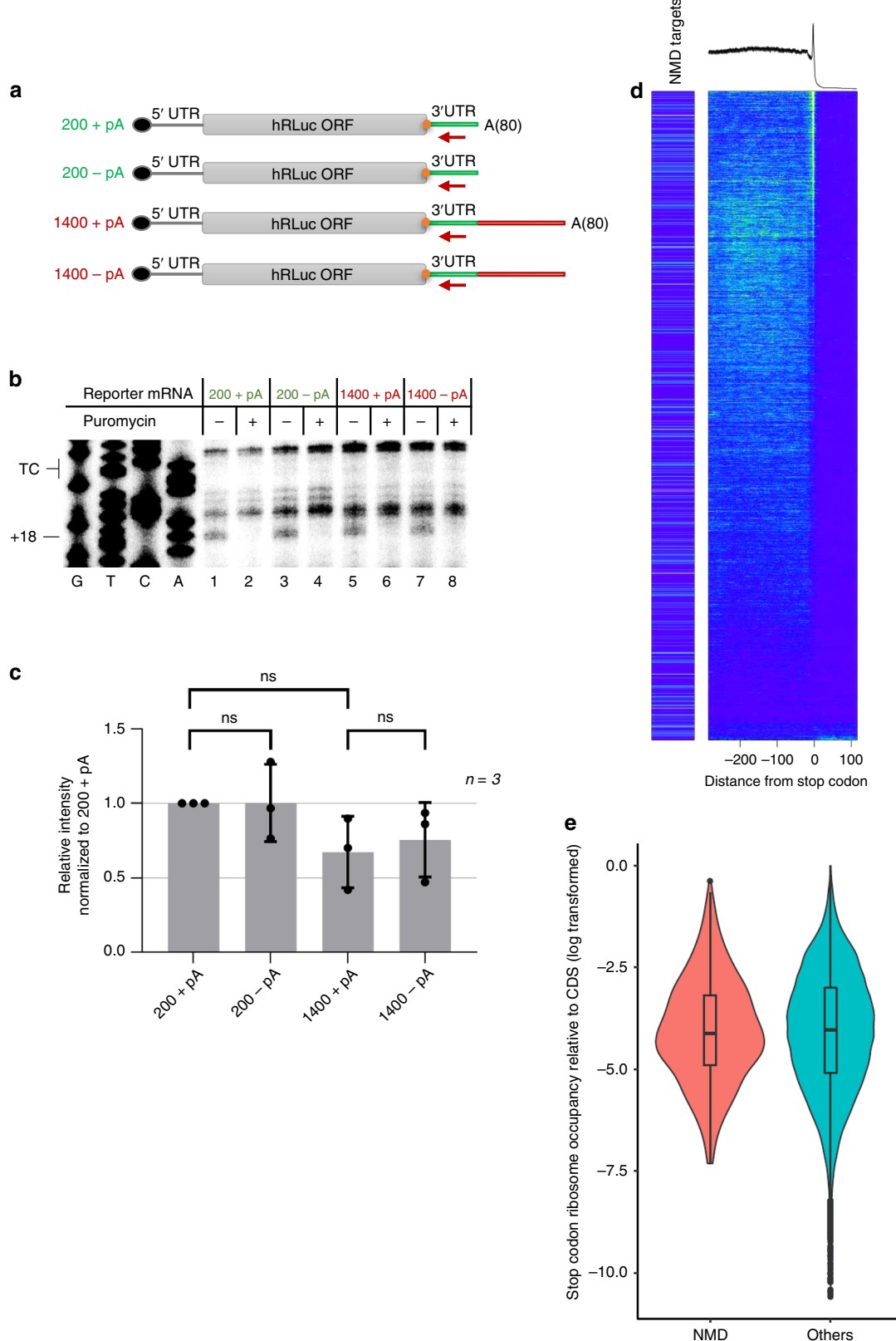

fully reconstituted eukaryotic systems allowed a detailed, stepwise functional interrogation of translation termination complexes and of the roles of release factors[26,27]. In this context, reconstituted translation is performed with isolated ribosomes that are supplemented with purified translation factors and aminoacylated tRNAs. The separate addition of each component allows tight

control of the system and its modulation (e.g. introducing nucleotide analogs or mutant versions of translation factors). On the other side, translation factor concentrations are far from physiological conditions and there is a requirement of intermediate purification steps of ribosome complexes, which limits the biological relevance of reconstituted translation systems.

**Fig. 5 Similar ribosome occupancy at the termination codon of NMD-sensitive and NMD-insensitive mRNAs. a** Scheme of in vitro synthesized RLuc reporter mRNAs. with (+pA) or without (−pA) a 80 nts long poly(A) tail. TC: orange dot. **b** Toeprint analysis with reporter mRNAs depicted in **a**. In vitro translation was performed in the presence (+) or absence (−) of puromycin. Sanger sequencing reactions were run in parallel (G,T,C,A). The TC position and the toeprint band corresponding to the ribosomes at the TC (+18) are indicated. **c** Relative quantification of +18 translation-dependent band from three individual experiments versus translation-independent mRNA signals normalized to 200 + pA conditions. Bars denote average values, dots depict the measurements of the three independent experiments, bars represent SD (Unpaired, two-sided statistical $t$-tests: 200-pA/p200 + A $p$ value = 0.983, 1400-pA/1400 + pA $p$ value = 0.701 1400 + pA/200 + pA $p$ value = 0,078). **d** (Top) Metagene analysis of ribosome-protected footprints from HeLa cells from three independent ribosome profiling experiments. Transcripts were aligned to the stop codon (0) and the mapped reads from 300 nucleotides upstream to 100 nucleotides downstream of the stop codon are shown. (Bottom) Heatmap of the ribosome-protected reads of all 60'000 transcripts considered for the metagene analysis. Transcripts were ordered according to their ribosome occupancy at the termination codon (transcripts with the highest number of reads at the stop codon at the top). NMD-sensitive transcripts are shown on the left of the heatmap panels. Each pixel line corresponds to the average of 10 transcripts. **e** Analysis of mean ribosome occupancy at the stop codon relative to mean ribosome occupancy in CDS, performed for NMD targets (identified as in **d**) and all other transcripts (Others). Violin plots showing the counts distribution of ribosome-derived reads mapping at the stop codon relative to ribosome-derived reads aligning to the CDS as an average of 3 biological replicates. Boxplots (middle) indicate the percentiles (5,25,50,75, and 95) of the ribosome occupancy values distribution with dots representing outliers. Percentage of reads relative to total reads are shown on the y-axis. Anova statistic test, $p$ value = 0.65. Source data are provided as a Source Data File.

While they are well suited to study mechanistic aspects of translation, they will fail to recapitulate many steps of translation regulation and processes more accessorily connected to translation. Since we are mainly interested in investigating the connection between translation and NMD, we opted for the development of a more physiologically relevant human-based in vitro system that does not require the isolation of specific ribosomal sub-complexes. This system allows all steps of translation of the in vitro transcribed reporter mRNA in the lysates to occur before toeprinting and detection of ribosomes at the TC. The system can be modulated, for example by knocking down specific factors or by the addition of recombinant proteins. We based our in vitro system on HeLa cells, because the majority of biochemical data available concerning human NMD derives from studies in this system. The development of an assay that allows monitoring of ribosome occupancy at the TC was an important achievement to test the idea that prolonged ribosome stalling at the TC is indeed the trigger for NMD.

Our in vitro toeprinting approach allows a direct comparison of ribosome occupancy at TCs between different reporter mRNAs translated in lysates of human cells. Extensive titration and optimization of different biochemical parameters (Supplementary Fig. 1) finally permitted that all steps of translation occur in the cell lysate, which was verified by the production of enzymatically active Rluc (Fig. 1b). Optimization of our system was focused on yielding reproducible translation-dependent bands in toeprint assays (Fig. 1c). To this end, a limited treatment of the lysates with micrococcal nuclease to reduce the number of ribosomes engaged in translating endogenous mRNAs, the titration of the primer distance from the TC and the optimization of $Mg^{2+}$ concentration to stabilize the ribosomes on the mRNA were crucial steps in establishing a successful protocol (Supplementary Fig. 1). Toeprint assays under the established conditions allow the reproducible detection of ribosomes at TCs and changes in the ribosome occupancy induced by stalling peptide sequences (Fig. 2) or depletion of translation termination factors (Fig. 3). The fact that our assay can detect ribosomes at the TC rather than at codons further upstream is in agreement with our ribosome profiling data that also showed an overall higher occupancy of ribosomes at the TC compared to the ORF. These two results agree with the notion that translational pauses at the termination codon is a common feature of translation[37].

As aforementioned, comparison between NMD-sensitive and NMD-insensitive mRNA reporters revealed a similar occupancy of ribosomes at the TC both in our in vitro toeprinting assay as well as by in vivo ribosome profiling (Fig. 5), suggesting that the

3′UTR length of an NMD-sensitive mRNA is not per se causing ribosome stalling at the TC. This data contrasts with two previous reports according to which increased ribosome density was observed on PTC-containing transcripts in yeast extracts and in rabbit reticulocyte lysates[18,19]. While it is conceivable that the difference between the yeast system and the human system reflects indeed a species difference, we speculate that the difference between the rabbit reticulocyte lysate and the human system might be attributed to toeprints originating from RNA cleavage in the reticulocyte lysate. We had initially also developed our toeprint assay using rabbit reticulocyte lysate and found that the toeprints we detected in this system almost exclusively represented co-translational RNA cleavages next to the ribosome (Supplementary Fig. 2), a feature that has been previously reported in reticulocyte lysates[43]. For this reason, we performed an additional control to monitor whether toeprints originate from ribosomes or RNA cleavage events by re-purifying the RNA after translation and prior to primer extension. A side-by-side comparison between these phenol-purified and untreated samples reveals whether toeprints derive from ribosomes or from cleaved RNAs.

The extent of increased ribosome occupancy at the termination codon may vary and stimulate different effects on mRNAs that are monitored by translation-dependent pathways. In cases of faulty mRNAs that are degraded due to a lack of a TC (nonstop decay) or due to a strong secondary structure or the absence of a cognate tRNA (no-go decay), the activating signal is the ribosome stalling during elongation. In this scenario, ribosome stalling marks a dead-end event and the rescue of an otherwise trapped ribosome is crucial (reviewed in refs. [17,44,45]). The term "stalling" is used broadly to describe increased ribosome density, without distinguishing between transient pauses and stable stalls. Even though a clear distinction between the two is still technically difficult, stalling as well as ribosome pausing have clearly been detectable by means of ribosome profiling or toeprinting[46–49] and both could be reproduced in our in vitro system. While we cannot exclude the possibility that NMD is associated with transient delays of translation at the termination codon that are too brief to be captured by means of toeprinting or ribosome profiling, our data clearly argue against the occurrence of stable ribosome stalling at TCs of NMD-sensitive mRNAs.

There is a growing body of evidence that NMD occurs stochastically depending on highly variable interactions that finally lead to the degradation of the mRNA, opposed to the traditional view of a linear, ordered, and irreversible pathway that leads to decay[7]. In yeast, NMD substrates were found to have an increased

rate of out-of-frame translation, accompanied by an overall decreased codon optimality or stretches of non-optimal codons compared to NMD-insensitive mRNAs[50]. This observation led to the idea that NMD is a constantly active mRNA surveillance pathway that monitors every mRNA throughout its life cycle[51]. The very recent observation that only a relatively small subset of termination events results in NMD using single-molecule kinetics of NMD-sensitive mRNA reporters[12] further supports the emerging view of a more complex mechanism of NMD activation and is inconsistent with stable ribosome stalling constituting the activation step of NMD. Collectively, these data and our results described herein warrant a critical revisiting and further testing of the prevailing NMD model that postulates ribosome stalling as the NMD-triggering signal.

## Methods

**Plasmids.** To create reporter constructs that differ in the length of the 3′UTR, a synthetic version of *Renilla* luciferase (Rluc) from the phRG-TK vector (Promega) was cloned into the pTRE-Tight vector (Clontech) via *Hind*III and *Xba*I, yielding pTRE-Tight-hRluc-200bp. For cloning of pTRE-Tight-hRluc-800bp 3′UTR, a 600 bp PCR product of the ampicillin resistance gene located in phRG-TK was generated using the primers 5′-AAT TTC TAG AAT TGT TGC CGG GAA GCT AGA GTA AG-3′ and 5′-AAT TTC TAG ATG AGT ATT CAA CAT TTC CGT GTC G-3′, cut with *Xba*I and inserted into the *Xba*I-linearized pTRE-Tight-hRluc-200bp 3′ UTR vector. A further elongation of the 3′UTR to 1400 bp (pTRE-TightRluc-1400bp 3′UTR) was achieved by a partial digestion of pTRE-Tight-hRluc-800bp 3′ UTR with XbaI and inserting the *Xba*I-digested 600 bp long PCR product a second time. The pTRE-Tight-hRluc constructs were subcloned into pCRII-TOPO (Invitrogen) according to the guidelines of the 'TOPO TA Cloning Kit Dual Promoter' (Invitrogen), resulting in pCRII-hRluc constructs with various 3′UTR lengths (pCRII-hRluc-200bp 3′UTR, pCRII-hRluc-800bp 3′UTR, pCRII-hRluc-1400bp 3′UTR). The reporter plasmids A, B and C are derivatives of pCRII-hRluc-800bp 3′UTR and were generated by site-directed mutagenesis using the primers 5′-CAA ATG TGG TAT GGC TGA TTA GAT CCT CTA GAA TTC CTG CTC-3′, 5′-CAA ATG TGG TAT GGC TGA TTA GAT CCT CAA GAA TTC CTG CTC-3′ and 5′-AAA TGT GGT ATG GCT GAT TGG ATC CTC AAG AAT TCC TGC-3′, respectively. We introduced the regulatory peptide of the human cytomegalovirus (hCMV) gp48 upstream open reading frame 2[30] into our 200 + pA reporter construct by fusion PCR yielding pCRII-hRLuc-200bp-hCMV SP-(A)80 and pCRII-hRLuc-200bp-hCMV CTRL-(A)80.

For expression tests in HeLa cells, we created the eukaryotic construct expressing the 200 + pA mRNA by amplifying the ORF of Rluc omitting the poly (A) signal that is harbored in the SV40-derived 3′UTR using the primers 5′-GGG CCC ATG GCT TCC AAG GTG TAC GA-3′ and 5′-GGT ACC AAC AAC AAC AAT TGC ATT CA-3′ and cut with *Apa*I and *Kpn*I to be inserted to equally treated pcDNA 3.1(−) vector yielding p200eu. To improve translation potential, the Kozak sequence[52] was optimized by site-directed mutagenesis using 5′-CTG GCT AGC GTT TAA ACG CCA CCA TGG CTT CCA AGG TGT-3′, yielding p200eukoz. For creating the eukaryotic construct for the expression of p1400 + pA, two regions of pCRII-hRluc_SV40_amp1200 (A)80 were amplified by fusion PCR using the primer pairs 5′-GGG CCC ATG GCT TCC AAG GTG TAC GA-3′/5′-TGG CGA TGA GAA CAA CAA CAA TTG CAT TCA-3′ and 5′-TGT TGT TGT TCT CAT CGC CAA TTG TTG CC-3′/5′-GGT ACC CTA GAT GAG TAT TC-3′. The fused product was then TOPO TA cloned into a pCRII-TOPO vector (Invitrogen) and ligated into the eukaryotic expression vector pcDNA 3.1 (−) using the restriction sites *Apa*I and *Kpn*I. The plasmids used for KD and rescue of UPF1 are described elsewhere[53].

**Cell lines and cell culture.** Lysate preparation and transfection experiments were performed using HeLa Tet-Off TCR-β ter 68 cells expressing a stably integrated TCR-β PTC+ (at position 68) minigene with an upstream tetracycline responsive element (TRE) and a minimal CMV promoter[10]. TRE promoter is regulated by a constitutively expressed Tet-Off advanced transactivator deriving from the parental cell line HeLa tTA-advanced clone 9 (Hela tetR clone 9)[54]. HeLa cells were cultured in Dulbecco's Modified Eagle Medium (DMEM) supplemented with FCS, Penicillin and Streptomycin (DMEM +/+) at 37 °C under 5% carbon dioxide atmosphere. The cells were grown in tissue culture flasks of variable sizes and kept in culture for no longer than one month. Passaging of cells was carried out by detachment adding Trypsin/EDTA solution (usually 1:10 (v/v) of the initial culture volume). The cell density was quantified by trypan staining and automated cell counting (Countess® Automated Cell Counter, Thermo Fisher scientific). HeLa cells were obtained from ATCC (CCL2).

**Preparation of translation-competent HeLa lysates.** Translation-competent HeLa lysates were prepared from approximately 80% confluent HeLa cell cultures ranging from $1 \times 10^7$ cells to $5 \times 10^8$ cells. Cells were washed with PBS pH 7.4 at room temperature (RT) and detached by trypsinization, resuspended in full growth medium, counted and pelleted (200 g, 4 °C for 5 min). The cell pellet was washed three times with ice cold PBS pH 7.4 and finally resuspended in ice-cold hypotonic lysis buffer [10 mM HEPES pH 7.3, 10 mM K-acetate, 500 µM Mg-acetate, 5 mM DTT and 1x protease inhibitor cocktail (biotool.com)] at a final concentration of $2 \times 10^8$ cells/ml. The suspension was incubated on ice for 10 min and cells were lysed by syringe treatment with a 1 ml syringe and a 27-gauge needle at 4 °C (cold room). The lysis process was monitored by trypan stain until more than 95% of the cells were lysed. The lysate was centrifuged at 13'000 g, 4 °C for 10 min and the supernatant was complemented to a final concentration of 1 mM CaCl₂ and 0.8 u/µl Micrococcal Nuclease (Thermo Fisher Scientific). The mixture was incubated at 20 °C for 10 min and then transferred on ice. The enzyme activity was quenched by addition of EGTA to a final concentration of 10 mM. Finally, the nuclease-treated HeLa lysate was aliquoted, snap frozen, and stored at −80 °C.

**In vitro transcription of reporter mRNAs.** A total of 4 µg of linearized pCRII vectors encoding the desired reporter mRNA downstream of a T7 promoter were mixed to yield an in vitro transcription reaction in 1x OPTIZYME™ Transcription Buffer (Thermo Fisher Scientific) to a final concentration of 40 ng/µl. The mixture further contained 1 mM of each ribonucleotide (rNTPs, New England Biolabs), 0.4 u/µl NxGen RNase inhibitor (Lucigen), 0.001 u/µl Pyrophosphatase (Thermo Fisher Scientific) and 5% (v/v) T7-RNA-polymerase (custom-made). The reaction was incubated at 37 °C for 1 h and then an equal quantity of T7-RNA polymerase was added for another 30 min. The mixture was then supplemented with TURBO DNase (Thermo Fisher Scientific) to a final concentration of 0.15 u/µl at 37 °C for 30 min. The transcribed mRNA was purified from the reaction using an acidic phenol-chloroform-isoamylalcohol (P.C.I) mixture followed by ethanol precipitation and two washes with 70% ethanol. The product was dissolved in disodium citrate buffer, pH 6.5, quantified by NanoDrop measurement and quality was assessed by agarose gel electrophoresis. Prior to capping, the RNA was incubated at 65 °C for 5 min and supplemented accordingly to yield a 100 µl reaction that consisted of 250 ng/µl RNA, 0.1 mM guanosine triphosphate (GTP, New England Biolabs), 0.1 mM S-adenosylmethionine (SAM, New England Biolabs), 2 u/µl NxGen RNase inhibitor (Lucigen), 0.1 u/µl vaccinia capping enzyme (VCE, New England Biolabs) in 1x Capping buffer (New England Biolabs). The capping reaction was carried out at 37 °C for 1 h and quenched by the addition of acidic P.C.I, followed by RNA purification. Capped mRNAs were quantified by Nano-Drop, 0.5 µg of RNA was analysed by agarose gel electrophoresis and aliquots were stored at −80 °C until use.

**In vitro translation.** For a typical in vitro translation reaction, an amount of lysate corresponding to $1.11 \times 10^6$ cell equivalents was used at a concentration of $8.88 \times 10^7$ cell equivalents/ml. The reaction was supplemented to a final concentration of 15 mM HEPES, pH 7.3, 0.3 mM MgCl₂, 24 mM KCl, 28 mM K-acetate, 6 mM creatine phosphate (Roche), 102 ng/µl creatine kinase (Roche), 0.4 mM amino acid mixture (Promega) and 1 u/µl NxGen RNase inhibitor (Lucigen). Control reactions contained 320 µg/ml puromycin (Santa Cruz Biotechnology). Before addition of mRNA, the supplemented lysate was incubated at 33 °C for 5 min. In vitro transcribed and capped mRNAs were pre-incubated at 65 °C for 5 min and cooled down on ice before addition to the pre-incubated translation reaction mixtures at a final concentration of 40 fmol/µl. Translation was performed at 33 °C for 50 min. To monitor the protein synthesis output, samples corresponding to $4.44 \times 10^5$ cell equivalents of the translation reaction were put on ice and mixed with 1x *Renilla*-Glo substrate (Promega) in *Renilla*-Glo (Promega) assay buffer on a white bottom 96 well plate. The plate was incubated at 30 °C for 10 min and the luminescence signal was measured three times using the TECAN infinite M100 Pro plate reader. Experiments using nuclease-treated Rabbit Reticulocyte Lysates (Promega) were performed by supplementing the reactions as above at 37 °C for 30 min following the manufacturer guidelines.

**Labeling of the toeprint primer and Sanger sequencing.** A labeling reaction contained 0.2 µM PAGE-purified primer (5′-TCA GGT TCA GGG GGA GGT G-3′), 0.39 u/µl T4 PNK (Thermo Scientific) and 0.59 µM [γ-³²P] ATP (Hartmann analytics). The mixture was incubated at 37 °C for 1 h and quenched by incubation at 68 °C for 10 min. The radiolabeled primer was separated from the free nucleotides using the Microspin G-25 Columns (GE Healthcare) according to the manufacturer's descriptions. Using the radiolabeled primer, a sequencing reaction was performed for each of the four nucleotides (G, T, C, A) using the same plasmid DNA templates that were used for in vitro transcription of the reporter mRNAs. The procedure was carried out using the USB Sequenase Version 2.0 DNA Sequencing Kit (Affymetrix) following the manufacturers guidelines.

**Toeprint assay.** After completion, translation reaction was incubated at 52 °C for 70 s and then immediately cooled down on ice. $2.66 \times 10^5$ cell equivalents of the stopped translation reaction were diluted to $2.72 \times 10^7$ cell equivalents/ml in a pre-cooled toeprint buffer containing (final concentrations given including the reverse transcriptase added at a later step) 37.0 mM Tris-HCl pH 7.3, 55.5 mM KCl, 5.0 mM MgCl2, 7.2 mM DTT, 0.2 mM dNTPs, 0.6 u/µl NxGen RNase inhibitor (Lucigen) and 20.3 nM radiolabeled primer. The mixture was pre-incubated at

37 °C for 5 min and AffinityScript Multiple Temperature Reverse Transcriptase (Agilent Technologies) was added to each toeprinting reaction at a final content of 10.2% (v/v, no unit definition available). Primer extension was carried out at 37 °C for 30 min. After completion, each reaction was diluted 1:11 in H$_2$O and placed on ice. The cDNA products were purified using the ChIP DNA Clean and concentrator (Zymo Research) according to the product manual. cDNA reactions were loaded on a 6% (v/v) polycrylamide gel containing 6.67 M urea in 1x TBE. Along with the samples, four Sanger sequencing reactions were loaded whereby the volumes were adjusted according to the radioactive counts of the cDNA samples (equal to counts). The gel was then run in 0.5x TBE at 27 W, fixed, dried and exposed overnight. The screen was scanned using the Typhoon FLA 9500 Laser scanner (pixel size 100 micrometer, sensitivity 100 V). Quantification of the toeprints was performed using ImageJ 1.52p where band intensities of translation-dependent toeprints were normalized to the overall intensity of translation-independent bands of the corresponding lane (background).

**siRNA-mediated ABCE1 depletion in HeLa cells**. The transfection mix added to the cells (grown to 40–60% confluency) consisted of 1:10 (v/v) of the culture medium and contained 25 nM siRNA (ABCE1 siRNA, 5′-GAG GAG AGU UGC AGA GAU UU dTdT-3′ or Negative Control siRNA, 5′-AGG UAG UGU AAU CGC CUU G dTdT-3′, Microsynth) and 0.25% (v/v) Lullaby transfection reagent (OZBiosciences) dissolved in Opti-MEM (Thermofisher). Before addition to the cells, the mix was incubated at RT for 20 min to allow complex formation. After 24 h of incubation, the cells were split to a higher format including a PBS pH 7.4 washing step. On the next day, the transfection was repeated under the same conditions and 24 h later cells were harvested and immediately processed to yield translation-competent lysates.

**Expression of Luc reporters in HeLa cells and NMD inhibition**. In order to express luciferase NMD reporters in vivo, HeLa cells (grown to 60–80% confluency) were transfected with the reporter constructs in pcDNA 3.1 (+) vectors (described above). Short hairpin RNA (shRNA)-mediated RNA interference (RNAi) degradation[55] was applied to knockdown UPF1 as described previously[35,53,55]. The transfection mix consisted of 20 ng/µl reporter plasmid, 20 ng/µl pSUPuro plasmid (1:1 mixture of pSUPuro UPF1 against two target sequences[53]) in Opti-MEM containing 3% (v/v) Dogtor (OZ Biosciences) transfection reagent. As a negative control, a pSUPuro plasmid containing a randomized target sequence was used (pSUPuro Scr). After 12 h the medium was replaced by DMEM +/+ containing 1.5 µg/ml Puromycin (Santa Cruz Biotechnology). The antibiotic selection was carried out for 48 h until the medium was replaced with DMEM +/+ to let the cells recover for approx. 24 h. To obtain a list of NMD-sensitive RNAs $3 \times 10^5$ HeLa cells per well were seeded in 6-well plates. Twenty-four hours later, the cells were transfected with 52 pmol of siRNA using Lullaby reagent (OZ Biosciences). After 48 h, the cells were re-transfected as before. Protein and total RNA were isolated after one additional day. The siRNA sequence 5′-GAUGCAGUUCCGCUCCAUU-3′ was used for targeting UPF1.

**Western blot**. To examine whether UPF1 or ABCE1 were depleted from HeLa cells and whether Rluc reporters are efficiently transfected or expressed after in vitro translation, cell lysates corresponding to $2 \times 10^5$ cells in the case of in vivo experiments or $4 \times 10^5$ per sample in the case of nuclease-treated lysates were analyzed by electrophoresis on 10 or 12% SDS-polyacrylamide gels and transferred onto nitrocellulose membranes (GE Healthcare Life Science). The proteins of interest were probed with antibodies against the following proteins: UPF1 (Bethyl A300-038A, 1:1000), ABCE1 (Abcam, ab185548, 1:1000), beta-actin (Sigma Aldrich A5060, 1:2000), Rluc (Thermo Fisher PA5-32210, 1:600), Tyr-Tubulin (Sigma T9028, 1:5000).

**RT-qPCR**. Approximately $2.7 \times 10^6$ HeLa cells were resuspended in 900 µl TRI reagent and total RNA was isolated with isopropanol as precipitation agent. The purified RNA was diluted in disodium citrate buffer, pH 6.5 and DNA was degraded using TURBO DNA-free™ Kit (Ambion, Thermo Fisher Scientific) following the manufacturers guidelines.

Reverse transcription reactions contained 50 ng/µl RNA, 15 ng/µl random hexamers, 1x AffinityScript RT buffer, 10 mM DTT, 0.4 mM dNTP mix (each), 1 u/µl NxGen RNase inhibitor (Lucigen) and 5% (v/v, no unit definition available) AffinityScript Multiple Temperature Reverse Transcriptase (Agilent). RNA was incubated at 65 °C for 5 min and was then left at RT for 10 min for primer annealing. The cDNA synthesis was carried out at 50 °C for 60 min and inactivated at 70 °C for 15 min. The cDNA was diluted with water to a final concentration of 8 ng/µl. For qPCR, each reaction consisted of 1.6 ng/µl cDNA and 0.5 µM of each primer specific for beta-actin (5′-TCC ATC ATG AAG TGT GAC GT-3′ and 5′-TAC TCC TGC TTG CTG ATC CAC-3′), Mini-TCRβ reporter (5′-AGT TGG CTT CCC TTT CTC AG-3′ 5′-CTT GGG TGG AGT CAC ATT TC-3′), Retinitis Pigmentosa 9 Pseudogene (RP9P) (5′-CAA GCG CCT GGA GTC CTT AA-3′ and 5′-AGG AGG TTT TTC ATA ACT CGT GAT CT-3′) or humanized Renilla luciferase (5′-CCC CGA GCA ACG CAA AC-3′ and 5′-GCA CGT TCA TTT GCT TGC A-3′) and in 1x Brilliant III Ultra-Fast SYBR® Green QPCR Master Mix

(Agilent Technologies). The reaction and fluorescence readout were performed in Rotor-Gene 6200 (Corbett Life Science) real-time system. Using the Rotor-Gene 6200 software (Corbett, version 1.7) threshold cycle values (ct-values) were set manually and the relative mRNA levels were subsequently calculated using the comparative CT method.

**Ribosome profiling**. Ribosome profiling was performed as in ref. [36]. Briefly, HeLa cells at 80% confluency were washed with ice-cold phosphate buffered saline (PBS) and flash-frozen in liquid nitrogen. Subsequently, cells were scraped and lysed in lysis buffer (20 mM Tris HCL pH7.4, 150 mM NaCl, 5 mM MgCl2, 1% Triton X-100, 1 mM DTT, 25 U/uL Turbo DNase, Turbo DNase buffer) on ice. Cells were then triturated ten times through a 27-gauge needle of a syringe and clarified by centrifugation. For ribosome profiling, lysates (about 4 U260) were subsequently treated with 200 U RNase I (Ambion) for 10 min at 23 °C and shaking at 300 rpm. The digestion was stopped by addition of 100 U SUPERase In RNase inhibitor (Ambion). Monosomes were separated on Illustra Micro-Spin S-400 HR gel filtration columns (GE Healthcare Life Science) as previously described[56]. TriReagent was added immediately to the eluates and samples were stored at −80 °C until further processing. RNA was isolated according to the TriReagent protocol and separated on 15% Novex polyacrylamide gels (Invitrogen). Ribosome footprints were excised between 26 and 34 nucleotide RNA size markers. After RNA isolation and purification, rRNAs were removed using the RiboZero kit (Illumina) according to manufacturer's datasheet. RNA was isolated from cleared lysates by addition of TriReagent as for the ribosome-protected fragments. Total RNA was used for library generation with the TruSeq Stranded mRNA Library Prep Kit (Illumina) according to the manufacturer's instructions. Libraries were sequenced on an Illumina HiSeq2500 generating 100 nt single-end reads. For compiling a list of NMD-sensitive transcripts we included the most abundantly expressed isoform under UPF1 KD treatment that originates from a previously identified list of NMD-sensitive genes[35]. Metagene analysis of ribosome-protected footprints from HeLa cells was performed from three independent ribosome profiling experiments. 639 NMD-sensitive transcripts were defined as the most abundant isoforms of previously identified NMD-sensitive genes[35] under UPF1 KD conditions. Mean ribosome occupancy at the stop codon was calculated relative to mean ribosome occupancy in CDS, performed for NMD targets (identified as in d) and all other transcripts (Others). Total counts of ribosome-derived reads mapping at the stop codon are plotted relative to the average of ribosome-derived reads aligning to the CDS for each biological replicate.

**Reporting summary**. Further information on research design is available in the Nature Research Reporting Summary linked to this article.

## Data availability
Ribo-sequencing data that support the findings of this study have been deposited in the Gene Expression Ominibus (GEO) under accession numbers: GSM4256659, GSM4256660, GSM4256661, GSM4256665, GSM4256666, GSM4256667 and total RNA sequencing data under UPF1 KD under accession numbers: GSM4407914, GSM4407915, GSM4407916. All data supporting the findings of this study are available within the paper and its Supplementary Information files. Source data for Figs. 1b, c, e, 2b, c, 3a–c, 4b, c, 5b, Supplementary Fig. 1b, c, d, Supplementary Figs. 2a, b, 3 were provided with the paper. All data is available from the corresponding author upon reasonable request. Source data are provided with this paper.

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

## Acknowledgements

We are grateful to Roland Beckmann (LMU Munich, Germany) and to Stefanie Metze for providing plasmids, to Lara Contu for proofreading of the manuscript, and to Asimina Gkratsou and Andrea Eberle for valuable discussions and advice. This work has been supported by the National Center of Competence in Research (NCCR) on RNA & Disease funded by the Swiss National Science Foundation (SNSF), by SNSF grants 31003A-162986 and 310030B-182831 to O.M., and by the canton of Bern (University intramural funding to O.M.).

## Author contributions

O.M. and E.D.K. conceived the project and designed the experiments. E.D.K., L.A.G., and O.M. designed translation experiments, E.D.K. and L.G. were involved in cloning and E.D.K. performed in vitro translation and toeprint reactions with the help of L.A.G. L.A.G. and E.D.K. performed in vivo experiments and prepared lysates. G.A. conducted the ribosome profiling and R.D. performed the bioinformatics analysis. All authors contributed to the final version of the manuscript.

## Competing interests

The authors declare no competing interests.
