## [Peer Review File · Nature Communications]

Reviewers' comments:

Reviewer #1 (Remarks to the Author):

The nonsense-mediated mRNA decay pathway has been intensively studied, but basic mechanistic questions about its execution remain unanswered. A model popular in the field for about 15 years has been that the NMD machinery targets mRNAs undergoing inefficient translation termination. This model is based on the idea that termination events at “premature” stop codons are slower than termination events at “normal” stop codons. One way in which termination efficiency is thought to be modulated and sensed in NMD is through competition between the NMD factor UPF1 and poly-A binding protein. PABPs have been described to enhance termination through interactions with the eukaryotic release factors; thus, in the absence of this interaction, the termination kinetics model posits that UPF1 is able to interact with eRFs and induce decay. Much of the evidence supporting this model is derived from in vitro assays of translation termination on NMD-sensitive and -insensitive mRNAs. In this important paper, Karousis and colleagues in the Muhlemann lab revisit the evidence for inefficient termination at NMD-inducing stop codons. They first describe the careful establishment of an in vitro ribosome toeprinting assay capable of distinguishing differential ribosome association at stop codons. With this system, they do not observe a difference in ribosome occupancy due to changes in 3'UTR length or the presence of a pA tail, contrary to prior models. Further, they show that ribosome toeprinting experiments in rabbit reticulocyte lysates may be confounded by NMD-independent RNA cleavage. In addition to the in vitro experiments presented, the authors also analyze ribosome profiling data, finding no evidence for differential termination efficiency at NMD-sensitive and -insensitive stop codons. While largely based on negative data, this work is a significant contribution to the ongoing debate over NMD mechanisms. Please see below for specific comments:

Major comments:

1. Figure 3B and Figure 5B: The authors use ABCE1 depletion in 3B to show that their assay is capable of detecting ribosomes stalled at termination codons, and then present data in 5B illustrating no difference in ribosome toeprints on NMD target and non-target mRNAs with and without polyA tails. The performance of the assay and impact of the negative result in 5B would be enhanced by quantification of the toeprinting assays in both figures.
2. Figure 5D shows that a previously defined set of NMD target transcripts and non-NMD target mRNAs have similar ribosome occupancy at their termination codons. This result is a little difficult to interpret without more complete information about how the transcripts were chosen for each set and how annotation ambiguity was dealt with. Were all transcripts arising from the NMD-sensitive genes identified in the Colombo et al paper used in the “NMD” set, or were genes divided into “NMD” and “non-NMD” transcript isoforms? If the former, the analysis may be confounded by multiple-counting of the same termination event or by assigning many normal termination events from non-NMD isoforms to the “NMD” category. If the latter, how were the annotations parsed to select which termination codon was used for analysis?
3. In general, the authors are careful to match their claims to the available data, but I think the title of

the paper overstates the findings somewhat. A title that conveys that the message of the paper is a lack of evidence for stalling at NMD-sensitive stop codons, rather than affirmative evidence that stalling does not occur would be preferable. Similarly, the last sentence of the abstract should be softened.

Minor comments:

1. The methods are generally thorough, but information about the quantity of in vitro transcribed mRNA added to the in vitro translation reactions should be included.
2. The description of the approach used to distinguish footprints from translation-dependent RNA cleavage could be clearer. I'm not sure whether the text at the bottom of page 3 is only meant to refer to the +18 band throughout or all of the bands observed, some of which still partially remain in the PC-extracted samples.
3. The blue and green colored dots used in Figure 1D and E are difficult to distinguish
4. Figure 2A legend: the sentence beginning, "The red arrow..." is prematurely terminated.
5. Figure 5C would benefit from a clearer illustration of the distribution of NMD targets across the Y axis.

Reviewer #2 (Remarks to the Author):

SUMMARY: In this study, Karousis et al. conduct a careful comparison of ribosome occupancy at the termination codon of NMD-sensitive versus insensitive substrates using in vitro and in vivo approaches. First, they optimize a toeprinting assay using translation-competent HeLa cell lysate to detect ribosome occupancy at the termination codon of an in vitro translated mRNA. Using this assay, they demonstrate an increased ribosome occupancy at the termination codon upon adding a hCMV uORF2 stalling peptide to their reporter, as well as upon depletion of the ribosome splitting factor ABCE1. They then use this assay to ask if an NMD-sensitive reporter mRNA (with a long 3' UTR) shows higher occupancy of ribosomes at the termination codon compared to the NMD-insensitive mRNA, and find this not to be the case. They complement this result with an analysis of a ribosome footprinting dataset, focusing on footprints at the termination codon, and show that there is no statistically significant difference in ribosome occupancy at the termination codon of NMD targets versus non-targets.

Taken together, the observations of Karousis et al. challenge the current view of NMD wherein stable ribosome stalling at the termination codon is a key trigger for NMD. While it is possible that there is transient pausing of the ribosome at the termination codon of NMD substrates at a time scale that is indistinguishable from normal termination by the assays used in this study, this is still an important distinction to be made from stable stalling of the terminating ribosomes on NMD substrates.

COMMENTS:

- Major: In some of the toe-printing gels (eg. Fig 5B), it looks as if the loading of RNA across the lanes is inconsistent. This paper would be strengthened by a more quantitative treatment of the toe-printing data including intensity measurement of the toe-print signal as well as normalizing this signal to that of the full-length mRNA. Given that every experiment was done in at least triplicates (as per Materials and Methods), a bar graph of the quantified signal could also be provided.
- Minor: The black and white color scheme of Figure 5C makes the plot hard to read. A more contrasting color or a different scaling that improves contrast would be preferable.
- Minor: A legend needs to be provided for Figure 5D that indicates what the different data points (filled square, empty square, and x's) are.

Reviewer #3 (Remarks to the Author):

Karousis et al. describe experiments intended to test a popular model for events occurring during translation of mRNAs that are targeted by the nonsense-mediated mRNA decay (NMD) pathway. Earlier work from Amrani et al. (Nature, 2004) and Peixeiro et al. (NAR, 2012) demonstrated, respectively, that yeast and human mRNAs harboring premature termination codons (PTCs) are not only *in vivo* substrates for NMD but, when translated *in vitro*, manifest ribosome pausing at their PTCs. The latter ribosome pausing events were detected in both papers by toeprinting, a method also known as primer extension inhibition. The earlier results were significant because they implied that termination at PTCs differed from normal termination and thus may have created a kinetic “window” for ribosomal association of factors such as Upf1 that are known to promote NMD. Here, Karousis et al. use a similar methodology, as well as ribosome profiling, to argue that the conclusions of the earlier studies cannot be confirmed and may, therefore, be wrong. While the subject of this study is important (because the mechanism underlying NMD remains unknown), the experimental evidence presented by the authors is quite thin and unconvincing. The following points address the shortcomings of the manuscript:

1. A major focus of this study is a set of *in vitro* translation experiments in HeLa cell lysates. As such, it is imperative that the authors characterize the translation activity of their lysates extensively and demonstrate that they are operating at peak efficiency. Hence, it was surprising that the authors chose very specific translation conditions without data supporting those choices. There was: a) no justification for the use of micrococcal nuclease treatment, b) no demonstration of a *bona fide* Mg²⁺ optimum, c) no mRNA concentration curve (actually no precise mRNA amount specified), d) no time course of protein synthesis, e) no demonstration that the synthetic mRNAs are fully capped or stable, and f) no assessment of the lysate's response to poly(A)⁺ or poly(A)⁻ mRNAs. There is an experiment showing that the authors' synthetic Rluc mRNA produces considerable luminescence when translated (Fig. 1B), but this data is meaningless in the absence of a comparison of that mRNA's specific activity in these lysates vs. others (e.g., purchased lysates). These points are not trivial because the work of Amrani et al. showed that the termination toeprints observed in yeast extracts prepared from wild-type cells could only be seen at PTCs and not at normal termination codons (NTCs). They argued that termination at

PTCs must be slower than that at NTCs, a point they substantiated by demonstrating that only extracts prepared from cells expressing a mutant release factor (eRF1) could yield toeprints at NTCs. The latter point is especially relevant here because the authors of this paper only look at toeprints from NTCs. The fact that they see such toeprints from their standard HeLa lysates implies that those lysates are most likely operating at low efficiency (i.e., the equivalent of the yeast lysates with defective eRF1).

2. The authors need to explain why:

a) All of their Rluc mRNA NTC toeprints are at +18, i.e., toeprints 18 nt 3' to the first nt of the nonsense codon. Comparable experiments with mammalian extracts (e.g., Pisarev et al., Mol Cell 2010) usually yield +15 toeprints.

b) The TC2 and TC3 mRNAs of Figure 1E have such diminished +18 toeprints. It would be expected that their intensities would be almost identical to that of TC1.

c) There was no toeprint from the CTR+pA mRNA (Fig. 2B). That mRNA should have yielded results comparable to their standard Rluc mRNA.

d) They considered the +18 band in the ABCE1 knock-down experiment of Fig. 3B to be of "higher intensity." There is so much variation in the +18 toeprints presented in this paper that a claim of "higher intensity" can only be justified by multiple repeat experiments and statistical assessments.

3. Fig. 4 demonstrates that the authors' standard mRNA (200 + pA) is "normal," i.e., not a substrate for NMD. Following on point #1 (above), this means that no toeprint should be detected from this mRNA in a normal extract, and that its presence implies that the extracts are translating inefficiently.

4. The experiments presented in Fig. 5C and D illustrate that the authors are missing key points of the model they hope to be testing. Why would they expect an accumulation of ribosomes at the normal termination codons of mRNAs known to be NMD substrates? These mRNAs are known to become NMD substrates by virtue of a variety of PTCs originating from uORFs, included introns, frameshift signals, etc., and if any ribosome pausing were to occur it would most likely happen at those PTCs, not at the NTCs. Further, the experiment suffers from a lack of statistical details, i.e., how many mRNAs were analyzed in the NMD "target" vs "other" groups, and what is the rationale for the specific use of Anova (vs. other tests)?

In short, this manuscript has created false tests of a model it seeks to refute and its negative results are thus not surprising or convincing.

Point-to-point response to the reviewers' comments

Reviewer #1

The nonsense-mediated mRNA decay pathway has been intensively studied, but basic mechanistic questions about its execution remain unanswered. A model popular in the field for about 15 years has been that the NMD machinery targets mRNAs undergoing inefficient translation termination. This model is based on the idea that termination events at “premature” stop codons are slower than termination events at “normal” stop codons. One way in which termination efficiency is thought to be modulated and sensed in NMD is through competition between the NMD factor UPF1 and poly-A binding protein. PABPs have been described to enhance termination through interactions with the eukaryotic release factors; thus, in the absence of this interaction, the termination kinetics model posits that UPF1 is able to interact with eRFs and induce decay. Much of the evidence supporting this model is derived from in vitro assays of translation termination on NMD-sensitive and -insensitive mRNAs. In this important paper, Karousis and colleagues in the Muhlemann lab revisit the evidence for inefficient termination at NMD-inducing stop codons. They first describe the careful establishment of an in vitro ribosome toeprinting assay capable of distinguishing differential ribosome association at stop codons. With this system, they do not observe a difference in ribosome occupancy due to changes in 3'UTR length or the presence of a pA tail, contrary to prior models. Further, they show that ribosome toeprinting experiments in rabbit reticulocyte lysates may be confounded by NMD-independent RNA cleavage. In addition to the in vitro experiments presented, the authors also analyze ribosome profiling data, finding no evidence for differential termination efficiency at NMD-sensitive and -insensitive stop codons. While largely based on negative data, this work is a significant contribution to the ongoing debate over NMD mechanisms. Please see below for specific comments:

Authors' response: We appreciate the fact that our work was perceived as a significant contribution to the ongoing discussion concerning NMD mechanism activation as well as the thoughtful and constructive comments by reviewer #1.

Major comments

1. Figure 3B and Figure 5B: The authors use ABCE1 depletion in 3B to show that their assay is capable of detecting ribosomes stalled at termination codons, and then present data in 5B illustrating no difference in ribosome toeprints on NMD target and non-target mRNAs with and without polyA tails. The performance of the assay and impact of the negative result in 5B would be enhanced by quantification of the toeprinting assays in both figures.

Response: As requested we performed a quantification of the toeprinting assays in both figures. The corresponding **figures 3 and 5** of the manuscript have been modified to include the analysis which verified a statistically significant increase of the translation-dependent band upon depletion of ABCE1 and statistically insignificant changes in ribosomal occupancy at the TC of mRNAs with or without a polyA tail as well as on mRNAs that are sensitive or insensitive to NMD.

We added the following remark in the section of material and methods: "*Quantification of the toeprints was performed using ImageJ 1.52p where band intensities of translation-dependent toeprints were divided by the overall intensity of translation-independent bands of the corresponding lane (background).*"

2. Figure 5D shows that a previously defined set of NMD target transcripts and non-NMD target mRNAs have similar ribosome occupancy at their termination codons. This result is a little difficult to interpret without more complete information about how the transcripts were chosen for each set and how annotation ambiguity was dealt with. Were all transcripts arising from the NMD-sensitive genes identified in the Colombo et al paper used in the "NMD" set, or were genes divided into "NMD" and "non-NMD" transcript isoforms? If the former, the analysis may be confounded by multiple-counting of the same termination event or by assigning many normal termination events from non-NMD isoforms to the "NMD" category. If the latter, how were the annotations parsed to select which termination codon was used for analysis?

Response: We apologize for the insufficiently detailed description of the analysis shown in Fig. 5D. Indeed, it is not possible to unambiguously assigning every transcript to either the category "NMD targets" or "Others", as this information does not exist for every transcript, and therefore we had to make some pragmatic decisions. In the initial analysis, all transcripts that belonged to a gene that was experimentally identified in the Colombo et al. paper as NMD target was assigned to the category "NMD targets". The reviewer is correct in stating that the inclusion of all transcripts of an NMD targeted gene into the "NMD targets" category leads to the inclusion of some actually NMD insensitive transcript into this category. To check whether this confounded our analysis, we repeated the analysis more stringently by assigning to the "NMD targets" category from the Colombo et al. NMD targeted gene list only the transcript of that gene that showed the highest expression under NMD inhibited conditions. The rationale for this selection is that the highest expressed transcript contributes the most to an observed differential expression at gene level and thus is most likely the NMD targeted transcript of an NMD targeted gene. This new analysis also showed no significant difference between "NMD targets" and "Others" with regard to the normalized ribosome density at the TC. In the revised version of the manuscript, we replaced the initial analysis (Fig. 5C and D) with this new, more stringent analysis and adjusted the text in the figure legend and results section, giving the previously missing information about how the transcripts were chosen for the two categories.

3. In general, the authors are careful to match their claims to the available data, but I think the title of the paper overstates the findings somewhat. A title that conveys that the message of the paper is a lack of evidence for stalling at NMD-sensitive stop codons, rather than affirmative evidence that stalling does not occur would be preferable. Similarly, the last sentence of the abstract should be softened.

Response: We tried indeed to be careful throughout the entire manuscript, including abstract and title, that our claims match to the available data. We believe that our data indeed support

that conclusion that there is no **stable** ribosomal stalling at NMD-sensitive stop codons, as our toeprint assay is able to report such stable stalling events (see e.g. Fig. 2), while we cannot exclude that we would miss the detection of less pronounced, more transient pausing events. We therefore find that the two statements "...ensues independently of stable ribosome stalling" (title) and "...is not accompanied by stable stalling of ribosomes at TCs" (abstract) accurately summarize our data and conclusions and hence we would be reluctant to change these sentences.

Minor comments

1. The methods are generally thorough, but information about the quantity of in vitro transcribed mRNA added to the in vitro translation reactions should be included.

Response: We are grateful for pointing out this unintended omission of important information in the Methods section. We now state the quantity of the *in vitro* transcribed mRNA used per translation reaction in Material and Methods. We have also included in Supplementary Figure 1 a titration of mRNA amounts to show that we are using an optimized mRNA concentration for our in vitro translations, as was requested by reviewer #3.

2. The description of the approach used to distinguish footprints from translation-dependent RNA cleavage could be clearer. I'm not sure whether the text at the bottom of page 3 is only meant to refer to the +18 band throughout or all of the bands observed, some of which still partially remain in the PC-extracted samples.

Response: We reformulated the corresponding text to clarify this point. Indeed, all translation-dependent bands in RRL derived from mRNA cleavage fragments, opposite to the +18 band in HeLa lysates that derived from terminating ribosomes.

3. The blue and green colored dots used in Figure 1D and E are difficult to distinguish

Response: We increased the overall size of the figure to ameliorate this point.

4. Figure 2A legend: the sentence beginning, "The red arrow..." is prematurely terminated.

Response: Thanks for spotting this mistake, the sentence was corrected.

5. Figure 5C would benefit from a clearer illustration of the distribution of NMD targets across the Y axis.

Response: We assume this comment pertains to Fig. 5D rather than 5C. We exchanged the box plot with a violin plot to provide a better and more intuitive display of the distribution of the data points.

Reviewer #2

In this study, Karousis et al. conduct a careful comparison of ribosome occupancy at the termination codon of NMD-sensitive versus insensitive substrates using in vitro and in vivo approaches. First, they optimize a toeprinting assay using translation-competent HeLa cell lysate to detect ribosome occupancy at the termination codon of an in vitro translated mRNA. Using this assay, they demonstrate an increased ribosome occupancy at the termination codon upon adding a hCMV uORF2 stalling peptide to their reporter, as well as upon depletion of the ribosome splitting factor ABCE1. They then use this assay to ask if an NMD-sensitive reporter mRNA (with a long 3' UTR) shows higher occupancy of ribosomes at the termination codon compared to the NMD-insensitive mRNA, and find this not to be the case. They complement this result with an analysis of a ribosome footprinting dataset, focusing on footprints at the termination codon, and show that there is no statistically significant difference in ribosome occupancy at the termination codon of NMD targets versus non-targets.

Taken together, the observations of Karousis et al. challenge the current view of NMD wherein stable ribosome stalling at the termination codon is a key trigger for NMD. While it is possible that there is transient pausing of the ribosome at the termination codon of NMD substrates at a time scale that is indistinguishable from normal termination by the assays used in this study, this is still an important distinction to be made from stable stalling of the terminating ribosomes on NMD substrates.

Response: We share the reviewer's view that while our assay may not be sensitive enough to report transient ribosome pausing or small kinetic differences in translation termination, it is still an important finding to show that there occurs no stable stalling of terminating ribosomes on the NMD substrates.

Major comments

In some of the toe-printing gels (eg. Fig 5B), it looks as if the loading of RNA across the lanes is inconsistent. This paper would be strengthened by a more quantitative treatment of the toe-printing data including intensity measurement of the toe-print signal as well as normalizing this signal to that of the full-length mRNA. Given that every experiment was done in at least triplicates (as per Materials and Methods), a bar graph of the quantified signal could also be provided.

Response: As requested, we performed the quantification of the toeprint assays for Figs. 3 and 5 and we provide the results in the form of bar plots as proposed in the updated version of the corresponding Figs. We added the following remark in the section of material and methods: "*Quantification of the toeprints was performed using ImageJ 1.52p where band intensities of translation-dependent toeprints were divided by the overall intensity of translation-independent bands of the corresponding lane (background).*"

Minor comments

- The black and white color scheme of Figure 5C makes the plot hard to read. A more contrasting color or a different scaling that improves contrast would be preferable.
Response: We replaced the black and white colour scheme of Fig. 5C by a coloured plot to improve clarity (new Fig. 5D).
- A legend needs to be provided for Figure 5D that indicates what the different data points (filled square, empty square, and x's) are.
Response: We have replaced the box plot of Fig. 5D (new Fig. 5E) with a violin plot to improve clarity and to provide a more intuitive way to assess the distribution of the data points.

Reviewer 3

Karousis et al. describe experiments intended to test a popular model for events occurring during translation of mRNAs that are targeted by the nonsense-mediated mRNA decay (NMD) pathway. Earlier work from Amrani et al. (Nature, 2004) and Peixeiro et al. (NAR, 2012) demonstrated, respectively, that yeast and human mRNAs harboring premature termination codons (PTCs) are not only *in vivo* substrates for NMD but, when translated *in vitro*, manifest ribosome pausing at their PTCs. The latter ribosome pausing events were detected in both papers by toeprinting, a method also known as primer extension inhibition. The earlier results were significant because they implied that termination at PTCs differed from normal termination and thus may have created a kinetic “window” for ribosomal association of factors such as Upf1 that are known to promote NMD. Here, Karousis et al. use a similar methodology, as well as ribosome profiling, to argue that the conclusions of the earlier studies cannot be confirmed and may, therefore, be wrong. While the subject of this study is important (because the mechanism underlying NMD remains unknown), the experimental evidence presented by the authors is quite thin and unconvincing.

Response: We regret that reviewer #3 found our evidence thin and unconvincing, but we appreciate the comments and feedback. We addressed below point-by-point the issues that reviewer #3 considered as shortcomings in our manuscript:

The following points address the shortcomings of the manuscript:

A major focus of this study is a set of *in vitro* translation experiments in HeLa cell lysates. As such, it is imperative that the authors characterize the translation activity of their lysates extensively and demonstrate that they are operating at peak efficiency. Hence, it was surprising that the authors chose very specific translation conditions without data supporting those choices. There was: a) no justification for the use of micrococcal nuclease treatment, b) no demonstration of a bona fide Mg²⁺ optimum, c) no mRNA concentration curve (actually no precise mRNA amount specified), d) no time course of protein synthesis, e) no demonstration that the synthetic mRNAs are fully capped or stable, and f) no assessment of the lysate's response to poly(A)⁺ or

poly(A)- mRNAs. There is an experiment showing that the authors' synthetic Rluc mRNA produces considerable luminescence when translated (Fig. 1B), but this data is meaningless in the absence of a comparison of that mRNA's specific activity in these lysates vs. others (e.g., purchased lysates). These points are not trivial because the work of Amrani et al. showed that the termination toeprints observed in yeast extracts prepared from wild-type cells could only be seen at PTCs and not at normal termination codons (NTCs). They argued that termination at PTCs must be slower than that at NTCs, a point they substantiated by demonstrating that only extracts prepared from cells expressing a mutant release factor (eRF1) could yield toeprints at NTCs. The latter point is especially relevant here because the authors of this paper only look at toeprints from NTCs. The fact that they see such toeprints from their standard HeLa lysates implies that those lysates are most likely operating at low efficiency (i.e., the equivalent of the yeast lysates with defective eRF1).

Response: We agree that the characterisation of translation activity is of utmost importance to set up an *in vitro* translation-based assay and most of the experiments that were suggested have been performed during the development of the assay. Since most of them are technical, and to our view a pre-requisite to present a new cell-free translation system, we did not include them in the first version of the manuscript. Responding to the criticism of the reviewer, we have now included some of the points in the updated version of **Supplementary Fig. 1** and the rest are embedded in this letter (i.e. for the reviewer's information, but not intended to be part of the paper). New Supplementary Fig. 1 now shows that **a)** we perform our assays at Mg²⁺ optimum, **b)** the chosen mRNA concentration of 40 fmol/μL is within the linear range of mRNA concentrations (i.e. not saturating), and **c)** that our *in vitro* translation system is still actively translating (i.e. in the linear phase of luc activity increase) 50 min after starting the reactions using 40 fmol/μL RNA and optimised Mg conditions. We apologize that we forgot to indicate in Material & Methods of the original submission how much mRNA was used for the translation reactions, this was an unintended omission. We modified the text in the results part as follows: "*After optimization of different parameters such as mRNA concentration, incubation time and titration of magnesium concentration to ensure efficient translation activity (Sup. Fig. 1), we utilized our in vitro translation-competent lysates to assess ribosome occupancy at the TC of reporter mRNAs by toeprinting assay*"

Regarding the stability of the *in vitro* transcribed and capped mRNAs, first we verified that they are capped by incorporation of radiolabeled GTP. 5 pmol of *in vitro*-synthesized 200-pA mRNA was incubated with or without Vaccinia capping enzyme under conditions described in material and methods. The experiment was performed two times in parallel, once with the addition of α³²P-GTP (1/10 of the concentration of total GTP) and once using only cold GTP. The RNA was run on a 1% agarose gel to assess its integrity and in parallel, the radiolabeled reactions were run on a 6% PAA TBE-Urea gel followed by autoradiography. In the experiment shown below, we compared side by side two different aliquots of capping enzyme (C2 and C3), because we suspected that C2 was not active. As shown in the autoradiography, we observed

a considerable incorporation of radiolabeled GTP in our capped mRNA when active VCE was included in the capping reaction mix (lane C3).

C1 C2 C3

6% PAA, Urea-TBE autoradiography

C1: No capping enzyme
C2: Defective capping enzyme
C3: Functional capping enzyme

Since capping is expected to increase translation efficiency (see Bergamini et al., *Cell-free translation systems*, 2002), we compared side by side translation of capped and uncapped 200+pA mRNAs in commercial MNase-treated rabbit reticulocyte lysate (RRL) and observed a 3-fold increase of luciferase activity, showing that our *in vitro* capping reaction indeed leads to a significant enhancement of translation efficiency.

To further assess the stability of the capped mRNAs in the HeLa lysates, we performed translation time course experiments followed by northern blotting to assess the stability of the reporter mRNAs

during the translation reaction. *In vitro* translation reactions were performed as described in Material & Methods for 200+pA and 800+pA reporter mRNAs. As shown in the following figure, both mRNAs remain intact for a time period of up to 120 min. This period of time is considerably longer than the duration of the reactions we report in the manuscript (50 min.).

Thus, the aforementioned experiments clearly showed that our *in vitro* transcribed mRNAs are adequately capped and are stable in our custom-made HeLa lysates for the entire duration of the *in vitro* translation reactions.

The reviewer also asked about our rationale for performing MNase treatments of the lysates. Applying a mild MNase treatment is strongly recommended in the literature for the preparation of translation-competent lysates¹ and we indeed observed in our translation system that an MNase treatment as described in Material & Methods improves the translation efficiency of our lysates. We titrated both the concentration of MNase as well as EGTA for quenching the nuclease reaction to identify optimal conditions, and here (Figure below) we show a side by side comparison of mock-treated HeLa lysates and MNase-treated lysate following the conditions specified in Material & Methods. The mock-treated samples underwent the same procedure, except that no MNase was added.

We observed a very reproducible increase of the *in vitro* translation activity by approx. 50% upon MNase treatment and therefore we included this step routinely in our final protocol.

The reviewer also considered it essential to compare the efficiency of our *in vitro* translation system with other, purchased lysates. To our knowledge, the only commercially available system based on mammalian cell lines is the Human Coupled IVT Kit by Thermo Scientific (Cat. Number 88881), which however can be used in conjunction with IRES-containing reporters. Even though our reporter mRNAs do not contain an IRES, we nevertheless compared the *in vitro* translation efficiency of our reporter mRNAs using our own lysates (Custom made) or following the manufacturer's guidelines for the IVT system (Commercial). As shown below, the efficiency of *in vitro* translation using the commercially available kit was more than 100-fold lower compared to our custom-made lysate (average of 3 measurements), mainly confirming that the commercial IVT system is not capable of translating efficiently normal mRNAs (i.e. transcripts that lack an IRES).

In summary, all of the above described data document that our *in vitro* translation assay is carefully optimized to perform with high efficiency. Noteworthy, similar optimizations of translation efficiency have not been shown for the experiments performed in yeast by Amrani *et al.* 2004, the paper that the reviewer #3 appears to use as the benchmark for judging our work.

2. The authors need to explain why:

a) All of their Rluc mRNA NTC toeprints are at +18, i.e., toeprints 18 nt 3' to the first nt of the nonsense codon. Comparable experiments with mammalian extracts (e.g., Pisarev et al., Mol Cell 2010) usually yield +15 toeprints.

Response: The reconstituted system by Pisarev *et al.* and our system are markedly different and the different toeprint positions observed relative to the stop codon may have at least two different explanations: 1.) the Pisarev system is *in vitro* reconstituted from recombinant or purified components and it almost certainly lacks several ribosome-associated factors that are present in our less well-defined cell lysates. The presence or absence of such factors may affect the ribosome conformation at the termination codon, which in turn might influence at which position on the mRNA the reverse transcriptase hits the ribosome and falls off. 2.) The use of different reverse transcriptases (MMLV versus AMV) could also explain the different positioning of the toeprints, due to different architectures and positioning of the catalytic domains. In any case, the following evidence convincingly demonstrates that our +18 toeprint band derives from ribosomes located at a TC: a) the +18 band disappears upon puromycin treatment (i.e. it's translation-dependent), b) the +18 band moves accordingly when the position of the TC, and c) the +18 band increased when we included a stalling peptide sequence (Fig. 2).

b) The TC2 and TC3 mRNAs of Figure 1E have such diminished +18 toeprints. It would be expected that their intensities would be almost identical to that of TC1.

Response: The explanation for this is technical. One parameter that has a strong influence on the sensitivity of the toeprint assay is the exact position of the radiolabelled primer and its relative distance to the position at which toeprints shall be monitored. The primer used throughout our study was designed to give optimal sensitivity at the position of the first TC (orange dot), but it is suboptimal and hence yielding weaker toeprints at the positions of the two TCs further down (blue and green dots). Therefore, no quantitative conclusions can be drawn from the toeprint intensities at the three different TCs. However, the fact that the +18 band disappeared and new bands appeared at positions exactly 18 nts downstream of the termination codons demonstrate that the signals correspond to terminating ribosomes.

c) There was no toeprint from the CTR+pA mRNA (Fig. 2B). That mRNA should have yielded results comparable to their standard Rluc mRNA.

Response: It is not clear to us on which assumptions the reviewer's expectations are based that the toeprint signals of two different assays performed with different mRNAs should be similar. The sequence contexts at the TCs are different between the standard Rluc mRNA (200+pA) and the CTR+pA mRNA, which we would expect to affect the kinetics of translation termination and hence the toeprint signals differently. The important conclusion from Fig. 2 is, that the intensity of the +18 band clearly increased in the presence of the stalling

peptide (SP+pA), which demonstrates the capacity of our toeprint system to comparatively monitor prolonged ribosome stalling at TCs.

d) They considered the +18 band in the ABCE1 knock-down experiment of Fig. 3B to be of “higher intensity.” There is so much variation in the +18 toeprints presented in this paper that a claim of “higher intensity” can only be justified by multiple repeat experiments and statistical assessments.

Response: As requested, we performed the quantifications and statistical assessment from four replicates of this experiment, and we have updated Fig. 3 accordingly. We added the following remark in the section of material and methods: “Quantification of the toeprints was performed using ImageJ 1.52p where band intensities of translation-dependent toeprints were divided by the overall intensity of translation-independent bands of the corresponding lane (background).”

3. Fig. 4 demonstrates that the authors’ standard mRNA (200 + pA) is “normal,” i.e., not a substrate for NMD. Following on point #1 (above), this means that no toeprint should be detected from this mRNA in a normal extract, and that its presence implies that the extracts are translating inefficiently.

Response: This point has been in-depth addressed above (point 1). The efficiency of our extracts in combination with a carefully established toeprint assay allowed the identification of terminating ribosomes on both NMD-sensitive and insensitive mRNAs. Whether a toeprint signal at the TC can be detected depends both on a) the efficiency of translation termination and on b) the overall sensitivity of the toeprinting assay. The fact that our assay can detect ribosomes at the TC rather than at codons further upstream is in agreement with our ribosome profiling data that also showed an overall higher occupancy of ribosomes at the TC compared to the ORF. These two results agree with the notion that translational pauses at the termination codon is a common feature of translation (Ingolia et al., *Cell*, 2011).

4. The experiments presented in Fig. 5C and D illustrate that the authors are missing key points of the model they hope to be testing. Why would they expect an accumulation of ribosomes at the normal termination codons of mRNAs known to be NMD substrates? These mRNAs are known to become NMD substrates by virtue of a variety of PTCs originating from uORFs, included introns, frameshift signals, etc., and if any ribosome pausing were to occur it would most likely happen at those PTCs, not at the NTCs. Further, the experiment suffers from a lack of statistical details, i.e., how many mRNAs were analyzed in the NMD “target” vs “other” groups, and what is the rationale for the specific use of Anova (vs. other tests)?

In short, this manuscript has created false tests of a model it seeks to refute and its negative results are thus not surprising or convincing.

Response: The reviewer's view that PTC-free "normal" mRNAs can become NMD substrates by virtue of a number of problems during translation is based on an interesting paper by Celik *et al.*, 2017. This is however so far the only publication reaching this conclusion and it is not clear whether the frequency of observed frameshifting in this study can quantitatively explain the corresponding reduction in mRNA levels, thus confirmation of these findings in other systems are warranted before referring to this working model as it was a commonly known and accepted fact ("These mRNAs are known to become NMD substrates..."). By contrast, there is ample published evidence that normal, protein-coding mRNAs can be targeted by NMD by virtue of their long 3'UTRs, and the rationale of our experiments is based on this NMD model. In accordance with this model, and shown in Fig. 4C, the 1400+pA mRNA reporter (which has a 3' UTR of 1400 nucleotides) is targeted by NMD. This NMD-sensitive transcript harbours none of the mentioned features (uORFs, included introns, known frameshift signals) and is identical in sequence with the NMD insensitive 200+pA reporters apart for the extended 3' UTR. We therefore with all respect reject the accusations that we should have applied false tests to investigate the current model that ribosome stalling at stop codons may lead to the activation of NMD. We maintain our conclusion that our experiments would have revealed any stable ribosome stalls at NMD-eliciting TCs if they existed.

Regarding the lack of statistical details given for the Fig 5C and D: Following the suggestions of reviewer #2, we repeated the ribosome profiling analysis by assigning to the "NMD targets" category from the Colombo *et al.* NMD targeted gene list only the transcript of that gene that showed the highest expression under NMD inhibited conditions. The rationale for this selection is that the highest expressed transcript contributes the most to an observed differential expression at gene level and thus is most likely the NMD targeted transcript of an NMD targeted gene. Importantly, as the initial analysis, this new and more stringent analysis showed no significant difference between "NMD targets" and "Others" with regard to the normalized ribosome density at the TC. The number of transcripts is (639) now mentioned on the updated version of the manuscript. A one way ANOVA test was used to compare the means of the relative stop codon ribosome occupancies between the "NMD target" and "Others" categories. With a p-value of 0.65, the means of the two categories show no significant difference.

REVIEWER COMMENTS

Reviewer #1 (Remarks to the Author):

The authors have comprehensively responded to my concerns. I just have one small correction: the Figure 5 references in the text were not updated to reflect the addition of toe-print quantification in 5c. The Fig. 5c reference in the text should be Fig. 5d, etc.

Reviewer #2 (Remarks to the Author):

The authors have adequately addressed my concerns with the manuscript via the changes made in the revised submission. I commend the authors on a well-conducted study that offers a strong counterpoint to the current thinking in the NMD field regarding ribosome stalling at the termination codon in NMD-sensitive transcripts.

Reviewer #3 (Remarks to the Author):

Karousis et al. have submitted a revised draft of a manuscript describing experiments intended to test a popular model for mechanistic aspects of nonsense-mediated mRNA decay (NMD). The revisions include: a) additional details about the in vitro translation system utilized for some experiments, b) quantitative analyses of toeprinting assays, c) a redefinition of the set of NMD substrates utilized for bioinformatic analyses of in vivo ribosome pausing, and d) improvements to the visual appearance of some figures and some minor issues in the text.

These revisions have addressed some, but not all of the requests from the three reviewers (further comments on those requests below). However, the new information provided has made it possible to determine more definitively whether the authors now have a convincing story worthy of publication in Nature Communications. I regret to say that this reviewer's opinion is "no." The basis for this assessment is as follows:

1. This is fundamentally a negative study, i.e., results were sought but could not be found. Negative studies by definition have a burden to prove that the test system(s) could have revealed the "positive results" if they were "real." Three major segments of the revised manuscript fail to meet these criteria:

a) In the toeprinting experiments of Figures 1 to 4 the authors sought to determine whether an mRNA that is targeted by NMD in vivo would manifest ribosomes stalled at premature termination codons in vitro, as had been seen in the earlier work of Amrani et al. (2004) and Peixero et al. (2006). A key question then is whether the in vitro translation system used for these analyses is capable of recognizing an NMD substrate as such, i.e., are there sufficient NMD factors present in the extract and are they active. In Amrani et al (2004), those authors showed that NMD was indeed functional in their extracts

because aberrant premature-termination-dependent toeprints were eliminated in extracts prepared from *upf1Δ* or *nmd2Δ* (*upf2Δ*) cells. Karousis et al. have now included details on the characterization of their extracts and these details provide no comparable assurance that UPF factors are active in their extracts. Accordingly, their inability to detect NMD-dependent paused/stalled ribosomes in vitro can easily be interpreted as a shortcoming of the test system, not the underlying NMD model.

b) In the bioinformatics analyses of Figure 5 the authors employed ribosome profiling to determine whether the set of mRNAs defined as endogenous in vivo NMD substrates manifested paused/stalled ribosomes at the normal termination codons located at the 3'-ends of their respective ORFs. This test should only apply to mRNAs that are NMD substrates because they possess extended 3'-UTRs, and 3'-UTR extension has been shown to make a normal termination codon behave as if it were a premature termination codon (PTC). Nevertheless, the authors carried out their analyses with all 639 NMD substrates (defined as the most abundant isoforms of transcripts that accumulate in response to UPF1 knock down). The complete set of NMD substrates will include mRNAs with extended 3'-UTRs, but also mRNAs with actual PTCs generated in different manners, including mRNAs that are transcripts of pseudogenes, mRNAs with retained introns, some mRNAs with uORFs, etc. The authors did not explain their decision to include all NMD substrates in this analysis, but it is clear that doing so makes it highly unlikely that statistically significant ribosome pausing at normal termination codons can be detected. The latter negative result is precisely what the authors saw.

c) The concern about negative results also applies to the authors' failure to detect a role for the mRNA poly(A) tail in the in vitro system. There are numerous reports in the literature demonstrating that an mRNA poly(A) tail enhances mRNA translational activity in vitro so, for the authors' negative result to be a significant finding they would have had to show that PABP and eRF3 levels (the factors most directly influenced by an mRNA poly(A) tail) in the system were normal, and that their synthetic mRNAs retained their poly(A) tails during in vitro translation. Otherwise, it's just an unsubstantiated negative result.

2. Other:

a) The authors were asked to provide evidence that their in vitro translation system was highly active, but failed to do so. The requested comparison of mRNA specific activities in an additional system was carried out in extracts that were IRES-dependent, i.e., extracts incapable of providing the requested test.

b) The explanation provided for the different toeprint strengths of PTCs A, B, and C should have been supported by experimental evidence.

c) The explanation for seeing +18 termination toeprint bands, as opposed to +15 or +16 seen by others using mammalian extracts, was highly speculative and not substantiated by results from the literature.

d) As noted by another reviewer, the title and conclusions of this paper are much too strong for the results obtained.

REVIEWER COMMENTS

Reviewer #1 (Remarks to the Author):

The authors have comprehensively responded to my concerns. I just have one small correction: the Figure 5 references in the text were not updated to reflect the addition of toe-print quantification in 5c. The Fig. 5c reference in the text should be Fig. 5d, etc.

Thanks for spotting this oversight, we have now updated and corrected the figure references to the panels in Figure 5.

Reviewer #2 (Remarks to the Author):

The authors have adequately addressed my concerns with the manuscript via the changes made in the revised submission. I commend the authors on a well-conducted study that offers a strong counterpoint to the current thinking in the NMD field regarding ribosome stalling at the termination codon in NMD-sensitive transcripts.

We thank reviewer #2 for appreciating the value that our study will bring to the field.

Reviewer #3 (Remarks to the Author):

Karousis et al. have submitted a revised draft of a manuscript describing experiments intended to test a popular model for mechanistic aspects of nonsense-mediated mRNA decay (NMD). The revisions include: a) additional details about the in vitro translation system utilized for some experiments, b) quantitative analyses of toeprinting assays, c) a redefinition of the set of NMD substrates utilized for bioinformatic analyses of in vivo ribosome pausing, and d) improvements to the visual appearance of some figures and some minor issues in the text.

These revisions have addressed some, but not all of the requests from the three reviewers (further comments on those requests below). However, the new information provided has made it possible to determine more definitively whether the authors now have a convincing story worthy of publication in Nature Communications. I regret to say that this reviewer's opinion is "no." The basis for this assessment is as follows:

1. This is fundamentally a negative study, i.e., results were sought but could not be found. Negative studies by definition have a burden to prove that the test system(s) could have revealed the "positive results" if they were "real." Three major segments of the revised manuscript fail to meet these criteria:

a) In the toeprinting experiments of Figures 1 to 4 the authors sought to determine whether an mRNA that is targeted by NMD in vivo would manifest ribosomes stalled at premature termination codons in vitro, as had been seen in the earlier work of Amrani et al. (2004) and Peixero et al. (2006). A key

question then is whether the in vitro translation system used for these analyses is capable of recognizing an NMD substrate as such, i.e., are there sufficient NMD factors present in the extract and are they active. In Amrani et al (2004), those authors showed that NMD was indeed functional in their extracts because aberrant premature-termination-dependent toeprints were eliminated in extracts prepared from upf1Δ or nmd2Δ (upf2Δ) cells. Karousis et al. have now included details on the characterization of their extracts and these details provide no comparable assurance that UPF factors are active in their extracts.

Accordingly, their inability to detect NMD-dependent paused/stalled ribosomes in vitro can easily be interpreted as a shortcoming of the test system, not the underlying NMD model.

Please see our response to the editor above. The reviewer is essentially requesting us to document that we have developed an active *in vitro* NMD system, something that many groups in the field including ours try to do since many years without success and which represents the “holy grail” in the NMD field. This request is completely unreasonable and has in our view only the purpose of trying to block acceptance of our findings because they contradict the previously published findings by Amrani et al. Notably, Amrani et al. did also not show “that NMD was indeed functional in their extracts”, they only showed that their complex and hardly interpretable banding pattern in their toeprints changed when extracts were prepared from yeast deficient of Upf1 or Upf2.

b) In the bioinformatics analyses of Figure 5 the authors employed ribosome profiling to determine whether the set of mRNAs defined as endogenous in vivo NMD substrates manifested paused/stalled ribosomes at the normal termination codons located at the 3'-ends of their respective ORFs. This test should only apply to mRNAs that are NMD substrates because they possess extended 3'-UTRs, and 3'-UTR extension has been shown to make a normal termination codon behave as if it were a premature termination codon (PTC). Nevertheless, the authors carried out their analyses with all 639 NMD substrates (defined as the most abundant isoforms of transcripts that accumulate in response to UPF1 knock down). The complete set of NMD substrates will include mRNAs with extended 3'-UTRs, but also mRNAs with actual PTCs generated in different manners, including mRNAs that are transcripts of pseudogenes, mRNAs with retained introns, some mRNAs with uORFs, etc. The authors did not explain their decision to include all NMD substrates in this analysis, but it is clear that doing so makes it highly unlikely that statistically significant ribosome pausing at normal termination codons can be detected. The latter negative result is precisely what the authors saw.

Please see our response to the editor above. Firstly, the analysis was performed on our group of high-confidence NMD targets identified in previous work (Colombo et al. 2017). The reviewer's assumption that these targets were identified simply based on a UPF1 knock down is wrong. Instead, these NMD targets represent genes significantly upregulated upon UPF1 and SMG6/SMG7 knockdowns and significantly downregulated by the respective rescue condition (see Colombo et al., 2017). Of those NMD targets, the stop codon belonging to the most upregulated transcripts under UPF1 KD was analyzed in the ribosome profiling. It is indeed an inherent limitation of the ribosome profiling technique that the ribosome protected fragments cannot be unambiguously assigned to one or another overlapping transcript isoform of a gene, which inevitably will result in some false negative and false positive assignments of reads. Nevertheless, we think that our approach is valid, since the most upregulated transcript isoform under depletion of NMD factors has the highest likelihood to be the transcript that renders the gene an NMD target. In any case, this selection criteria for the ribosome profiling analysis is certainly more reasonable than the criteria suggested by reviewer #3 for the following reason: Opposite to yeast, where long 3'UTRs comprise the main NMD-triggering feature, the 3'UTR length in human cells has only very limited predictive power for whether an mRNA will trigger NMD (see e.g. Colombo et al, RNA 2017; Ge et al., eLife 2016). The best characterized and most frequent NMD-triggering feature (with a predictive power of about 70%) is in human cells the presence of an intron >50 nucleotides downstream of the stop codon. Thus, restricting our ribo-seq analysis on the relatively small subset of NMD targets with long 3'UTRs would therefore exclude the bulk of human endogenous NMD-sensitive mRNAs and so primarily reduce the statistical power of the analysis. Analyzing a much smaller subgroup would only be meaningful, if this subgroup consisted of a higher fraction of true direct NMD-targets, which is clearly not the case here, because the likelihood that a transcript with a long 3'UTR is a direct NMD target is in fact much smaller than the likelihood that a transcript with an intron >50

nucleotides downstream of the stop codon is an true NMD target. Thus, even though technically feasible, the requested analysis would yield a less reliable result, because it would be conducted on a small subgroup of transcripts, of which a smaller fraction would represent direct NMD targets. We therefore see no benefit in this approach and maintain our view that the currently presented analysis yields the results with the highest possible reliability and statistical robustness.

c) The concern about negative results also applies to the authors' failure to detect a role for the mRNA poly(A) tail in the in vitro system. There are numerous reports in the literature demonstrating that an mRNA poly(A) tail enhances mRNA translational activity in vitro so, for the authors' negative result to be a significant finding they would have had to show that PABP and eRF3 levels (the factors most directly influenced by an mRNA poly(A) tail) in the system were normal, and that their synthetic mRNAs retained their poly(A) tails during in vitro translation. Otherwise, it's just an unsubstantiated negative result.

We are very much aware and also concerned that we present here to a large extent negative results that by definition can never be fully conclusive and we were therefore careful to not overinterpret them. Nevertheless, the fact that both *in vitro* (toeprints) and *in vivo* (ribosome profiling) approaches did not yield any evidence for prolonged ribosome stalling at stop codons that elicit NMD is an important finding for the RNA turnover community, because it sets a counterpoint to the current thinking in the field (as pointed out by reviewer #2) and warrants re-visiting this model of ribosome stalling at PTCs, which so far is essentially based on one single paper (Amrani et al.). Importantly, we do not conclude that this model is wrong, we only say that in the light of our results, it should not be taken for granted. While we cannot exclude that both our assays might not be sensitive enough to detect transient ribosome stalling at NMD-eliciting stop codons, we can exclude the occurrence of stable stalling, since we show with respective controls that we can detect stable stalling in both our assays (see Figs. 2, 3 and our manuscript by Annibaldis et al., bioRxiv 870097 (2019). doi:10.1101/870097).

2. Other:

a) The authors were asked to provide evidence that their in vitro translation system was highly active, but failed to do so. The requested comparison of mRNA specific activities in an additional system was carried out in extracts that were IRES-dependent, i.e., extracts incapable of providing the requested test.

To the best of our knowledge, there is no commercially available IRES-independent *in vitro* translation system from mammalian cells that could be used as a benchmark for the performance of our system. However, the additional test shown in Suppl. Figs. 1a-c document that under our conditions, the translation activity is high but still not saturated/limited.

b) The explanation provided for the different toeprint strengths of PTCs A, B, and C should have been supported by experimental evidence.

This experiment was performed to assess whether the +18 that was observed in our toeprints depends on translation and has a qualitative rather than a quantitative character. The fact that this band disappears when TC1 is mutated and that it moves to a position +18 of the corresponding termination codons provides strong evidence that we observe translation termination-dependent bands. That the distance between the primer and the stalled ribosome affect the intensity of the toeprint band is well known and has e.g. also been reported by Peixeiro et al. (2012). Since we use the same primer to detect toeprints of termination codons at different positions in our experiment, different signal intensities are to be expected. Hence we do not see a reason for additional experiments.

c) The explanation for seeing +18 termination toeprint bands, as opposed to +15 or +16 seen by others using mammalian extracts, was highly speculative and not substantiated by results from the literature. As described in our previous response to reviewer's comments, there are several possible explanations for this discrepancy. Some results from the literature that substantiate these points are the following: In yeast, translation-dependent toeprints have been observed in a range of 6-19 nts downstream of nucleotides occupied by the P or the A site of stalled ribosomes (Sachs et al., 2002) and +18 toeprints have been observed downstream of the initiation codon in Rabbit Reticulocyte lysate (Peixeiro et al.,

2012). Therefore the position of our translation-dependent toeprints 18 nucleotides downstream of the termination codon may not agree with observations that were made using the unique mammalian reconstituted translation system, but it definitely falls within the range that has been observed in eukaryotic translation-dependent toeprints.

d) As noted by another reviewer, the title and conclusions of this paper are much too strong for the results obtained.

As mentioned in our previous response to reviewer's comments, we conclude from our work that there happens no stable stalling during translation termination on termination codons of NMD-sensitive mRNAs compared to NMD-insensitive transcripts, while we cannot exclude subtle differences in ribosome residence time at NMD-triggering termination codons. This is also what we state in the title and the abstract and we therefore see no reason to change the title or the abstract.